# Multi-Instance Causal Representation Learning for Instance Label Prediction and Out-of-Distribution Generalization

**Weijia Zhang**[1][*]**, Xuanhui Zhang**[2]**, Han-Wen Deng**[1]**, Min-Ling Zhang**[1]

[1]School of Computer Science and Engineering, Southeast University, Nanjing 210096, China
[2] School of Information Management, Nanjing University, Nanjing 210023, China
zhangwj@seu.edu.cn, zhangxhdo@163.com, {denghw,zhangml}@seu.edu.cn

## Abstract

Multi-instance learning (MIL) deals with objects represented as bags of instances and can predict instance labels from bag-level supervision. However, significant performance gaps exist between instance-level MIL algorithms and supervised learners since the instance labels are unavailable in MIL. Most existing MIL algorithms tackle the problem by treating multi-instance bags as harmful ambiguities and predicting instance labels by reducing the supervision inexactness. This work studies MIL from a new perspective by considering bags as auxiliary information, and utilize it to identify instance-level causal representations from bag-level weak supervision. We propose the CausalMIL algorithm, which not only excels at instance label prediction but also provides robustness to distribution change by synergistically integrating MIL with identifiable variational autoencoder. Our approach is based on a practical and general assumption: the prior distribution over the instance latent representations belongs to the non-factorized exponential family conditioning on the multi-instance bags. Experiments on synthetic and real-world datasets demonstrate that our approach significantly outperforms various baselines on instance label prediction and out-of-distribution generalization tasks.

## 1   Introduction

Supervised learning has achieved great success in many applications. However, an important limitation of existing supervised learning algorithms is that they model complex objects as a single feature vector and thus cannot make fine-grained predictions without the corresponding level of supervision, e.g., localizing the region of interest within an image from coarse-grained image-level supervision. Unfortunately, acquiring fine-grained labels is often not only a tedious task but also prohibitively expensive, especially for applications that require a high level of domain-specific expertise such as drug activity prediction [8] and medical image classification [24].

Multi-Instance Learning (MIL) [8] is a weakly supervised learning paradigm originally proposed for drug activity prediction, where the task is to predict whether a molecule is suitable for binding to a target receptor. Since a molecule can take many low-energy conformations and its suitability for making drugs depends on some specific but unknown conformations, objects in MIL are represented by groups of instances called *bags* where each instance is described by its own feature vector, instead of represented by a single feature vector as in standard supervised learning. Because only molecule-level drug binding suitability is known to human experts, MIL algorithms are only coarsely supervised at the bag-level, while fine-grained instance labels are unknown.

---

[*]Corresponding author

36th Conference on Neural Information Processing Systems (NeurIPS 2022).

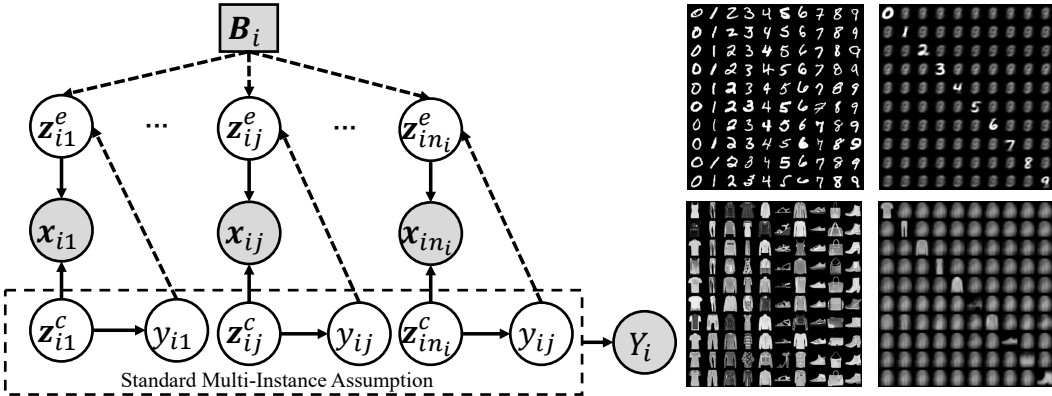

Figure 1: (Left) Our proposed graphical model. Shaded nodes denote observed variables and white nodes represent the latent ones. Dashed lines denote edges that may exist depending on the bags, while solid lines indicate relationships that are invariant with respect to the bags. (Right) Multi-instance bags of MNIST and FashionMNIST images and their reconstructions from $z^c$ (each class per row) inferred by CausalMIL.

Utilizing the bag-level supervision, the prediction tasks of MIL are two-fold: predicting the bag labels, e.g., whether a molecule is suitable for making drugs or a human organ contains cancerous cells; and predicting the instance labels, e.g., which specific molecular conformations are suitable or which particular cells are cancerous. Although theoretical results have demonstrated the feasibility of learning accurate instance concepts from bag labels [9], the empirical performances of weakly supervised instance label predictions are far behind their fully supervised counterparts.

The inexactness in MIL supervision causes the main difficulty of instance label prediction [42]. Under the standard multi-instance assumption [11], the bag labels only inform the learner that instances from negative bags are negative, while the exact labels for instances in the positive bags are unknown. Most previous instance-level MIL algorithms solve the above problem by disambiguating the bag labels, i.e., finding the most positive instance from positive bags [28, 35] or utilizing the attention mechanism to infer how the different instances contribute to the bag label [15, 31, 29].

In this work, we propose to utilize the information hidden in the multi-instance bags for identifying the latent factors that generate the observed instances, and utilize the bag labels for encouraging disentanglement among the latent factors. Specifically, we consider each observed instance as generated from instance-specific causal factors $z_{ij}^c$ and bag-inherited non-causal factors $z_{ij}^e$ as depicted in Figure 1 (left). The instance-specific factors $z_{ij}^c$ are responsible for capturing information that is causal to the instance labels, while the bag-inherited factors $z_{ij}^e$ capture spurious information that may be inherited from the bags and correlates with the instance labels. The bag labels are considered as the effects of their instance labels according to the standard multi-instance assumption.

By identifying and disentangling the latent factors, our model brings two advantages over existing MIL algorithms. Firstly, as the identified latent factors capture semantic meaningful causal representations are of much lower dimensions than the observed instances, their identification significantly reduces the difficulties of inferring instance labels from bag supervision. Secondly, since only the instance-specific factors $z_{ij}^c$ are causes to the instance labels $y_{ij}$, disentangling $z_{ij}^c$ from $z_{ij}^e$ remove the spurious correlations between $B_i$ and $y_{ij}$. Therefore, identification and disentanglement of $z_{ij}^c$ is not only beneficial for instance label predictions since $z_{ij}^e$ will vary among bags, but is also useful for out-of-distribution (OOD) generalization since the factors that are the effects of instances labels are excluded from the predictive model [4, 1, 23]. It is worth noting that although $B_i$ and $y_{ij}$ are unconditionally independent, they become conditionally dependent because previous algorithms that do not identify or disentangle the latent factors implicitly condition on either $x_{ij}$ or $z_{ij}^e$ (i.e., the collider or descendants of the collider in a v-structure [26]).

Our identifiability result builds upon the recent identifiable variational autoencoder (iVAE) [17], but extends to a more flexible non-factorized prior distribution conditioning on the bag information instead of the factorized prior conditioning on a single instance assumed in the original iVAE. Furthermore, we propose to utilize a permutation invariant set transformation network for aggregating information

from instances within the same bag to facilitate identifiability. Because the set transformation network universally approximates any set function [39], instances within the same bag are not only treated as sources of supervision inexactness but also effectively contribute to identifiability via conditioning the prior distribution. We summarize our key contribution as follows:

- We propose a novel CausalMIL algorithm that utilizes the bag information with permutation invariant set transformation networks to theoretical guarantee the identifiability of latent factors under non-factorized conditional exponential family priors;

- We propose a general graphical model covering many real-world MIL applications, and show that the causal and non-causal factors of the instance labels can be disentangled from the identified latents based on bag-level supervision.

- Using a variety of datasets, we show that CausalMIL significantly outperforms existing MIL algorithms on instance label prediction tasks, and achieves better out-of-distribution generalization capabilities when compared to supervised invariant algorithms.

## 2 Preliminaries

### 2.1 Notations

Let $\mathcal{X} = \mathbb{R}^m$ denote the instance space and $\mathcal{Y} = \{0, 1\}$ denote the label space. The training data contains samples organized in $n$ training bags $\mathcal{B} = \{\mathbf{B}_1, \cdots, \mathbf{B}_i, \cdots, \mathbf{B}_n\}$, where each bag is a set that contains different numbers of $n_i$ instances, i.e., $\mathbf{B}_i = \{\boldsymbol{x}_{i1}, \cdots, \boldsymbol{x}_{ij}, \cdots, \boldsymbol{x}_{in_i}\}$ with $\boldsymbol{x}_{ij} \in \mathcal{X}$. In the rest of this paper, we will also use $\mathbf{B}_i$ to denote the auxiliary information contained in the bag by abuse of notation. The learning algorithm is provided with bags $\mathbf{B}_i$ and their associated bag labels $Y_i \in \mathcal{Y}$ during training. Although there exists instance labels $y_{ij} \in \mathcal{Y}$ for each $\boldsymbol{x}_{ij}$, the instance labels are *unknown* to the learners. We follow the standard multi-instance assumption [11] which assumes that positive bags contain at least one positive instance, and negative bags contain only negative instances. It is worth noting that the assumption only requires bags to be i.i.d., the instances within the same bag can be dependent.

### 2.2 VAE and Identifiability

We briefly introduce Variational Autoencoder (VAE) and its identifiability results. VAE can be considered as the combination of a generative latent variable model and an associated inference model, where both models are parameterized by neural networks [18]. Specifically, VAE learns the joint distribution $p_{\boldsymbol{\theta}}(\boldsymbol{x}, \boldsymbol{z}) = p_{\boldsymbol{\theta}}(\boldsymbol{x}|\boldsymbol{z})p_{\boldsymbol{\theta}}(\boldsymbol{z})$ where $p_{\boldsymbol{\theta}}(\boldsymbol{x}|\boldsymbol{z})$ is the conditional distribution of observing $\boldsymbol{x}$ given $\boldsymbol{z}$, $\boldsymbol{\theta}$ is the set of generative parameters, and $p_{\boldsymbol{\theta}}(\boldsymbol{z}) = \Pi_{i=1}^{d} p_{\boldsymbol{\theta}}(z_i)$ is the factorized prior distribution of the latents. By introducing an inference model $q_{\boldsymbol{\phi}}(\boldsymbol{z}|\boldsymbol{x})$, the set of parameters $\boldsymbol{\phi}$ and $\boldsymbol{\theta}$ can be jointly optimized through maximizing the evidence lower bound (ELBO) on the marginal likelihood $p_{\boldsymbol{\theta}}(\boldsymbol{x})$:

$$\begin{aligned}
\mathcal{L} &= \mathbb{E}_{q_{\boldsymbol{\phi}}(\boldsymbol{z}|\boldsymbol{x})}[\log p_{\boldsymbol{\theta}}(\boldsymbol{x}|\boldsymbol{z})] - D_{\mathrm{KL}}(q_{\boldsymbol{\phi}}(\boldsymbol{z}|\boldsymbol{x})||p(\boldsymbol{z})) \\
&= \log p_{\boldsymbol{\theta}}(\boldsymbol{x}) - D_{\mathrm{KL}}(q_{\boldsymbol{\phi}}(\boldsymbol{z}|\boldsymbol{x})||p_{\boldsymbol{\theta}}(\boldsymbol{z}|\boldsymbol{x})) \leq \log p_{\boldsymbol{\theta}}(\boldsymbol{x}),
\end{aligned} \tag{1}$$

where $D_{\mathrm{KL}}$ denotes the KL-divergence between the approximation and the true posterior, and $\mathcal{L}$ is a lower bound of the marginal likelihood $p_{\boldsymbol{\theta}}(\boldsymbol{x})$ because of the non-negativity of the KL-divergence.

Recently, it has been shown that VAEs with unconditional prior distributions $p_{\boldsymbol{\theta}}(\boldsymbol{z})$ are not identifiable [22], but the latent factors $\boldsymbol{z}$ can be identified with a conditionally factorized prior distribution $p_{\boldsymbol{\theta}}(\boldsymbol{x}|\boldsymbol{u})$ over the latent variables to break the symmetry, where $\boldsymbol{u}$ is an additionally observed variable [17].

## 3 Methods

### 3.1 CausalMIL

We now formally describe our graphical model depicted in Figure 1 (left). For any given bag $\boldsymbol{B}_i$, we model its observed instances $\boldsymbol{x}_{ij}$ as generated from unknown latent factors $\boldsymbol{z}_{ij} = (\boldsymbol{z}_{ij}^c, \boldsymbol{z}_{ij}^e) \in \mathbb{R}^d$ with $d \ll m$. We let $\boldsymbol{z}_{ij}^c$ correspond to the causes of the instance label $y_{ij}$, i.e., the line stroke that

makes a digit represent the Arabic number '1' in Figure 1 (right). In contrast $z_{ij}^e$ captures other factors related to the observed $x$ but are not causes of the instance label, i.e., the angle, thickness, and color of the line stroke that are not causal to '1'. The solid lines of the graphical model indicate that the causal relationships are invariant across all bags, and the dashed lines denote that their relationships may vary or even be absent across different bags. The bag label $Y_i$ is determined by the labels of its instances $y_{ij}$ according to the standard multi-instance assumption. To sum up, we posit that the graphical structure in Figure 1 should satisfy the following assumptions:

**Assumption 1.** *(a) The bag labels are determined by its instances according to the standard multi-instance assumption; (b) the causal graph in Figure 1 is a Directed Acyclic Graph (DAG); (c) given the latent factors $z_{ij}$, the observed instances are independent of the labels and the bags: $x_{ij} \perp\!\!\!\perp y_{ij}, Y_i, B_i | z_{ij}$, i.e., the generation mechanism $p(x_{ij}|z_{ij})$ is invariant across bags; (d) $y_{ij} \perp\!\!\!\perp B_i | z_{ij}^c$, i.e., the mechanism between the causal factors $z_{ij}^c$ and $y_{ij}$ is invariant across bags.*

We now discuss the practicality of Assumption 1. Firstly, the standard multi-instance assumption commonly used in a wide range of MIL applications such as medical image analysis [20], fine-grained sentiment analysis [3], and sound event detection [36]. Secondly, modeling the causal structures as DAGs are common in causal discovery [26, 27]. To the best of our knowledge, most of the causality-based machine learning literature assumes the causal structure to be acyclic. Thirdly, it also makes sense that generating mechanism from latent factors to observed instances $p(x_{ij}|z_{ij})$ is invariant since causal mechanisms are considered to be stable across environments [30]. Otherwise, it would be impossible to infer $z$ from $x$ for any test instances. Lastly, the fourth assumption is not only commonly adopted in invariant and causal representation learning literature [4, 1, 23], but also suitable for MIL applications. For example, when diagnosing whether a cell from a tissue is cancerous, the causal mechanism of diagnosing a cell should be invariant across bags, while the non-causal factors caused by the patient or equipment differences may often change [40].

From now on we will drop the subscript to avoid cluttering the notations. In the following discussion we will show that when the underlying data generating mechanism satisfies Assumption 1, the latent variables $z$ can be identified up to permutation and affine transformations if the conditional prior distribution $p(z|B, y)$ belongs to a general exponential family distribution:

**Assumption 2.** *The prior distribution of the latent factors $p(z|B, y)$ follows a general exponential family with its parameter specified by an arbitrary function $\lambda(B, y)$ and sufficient statistics $T(z) = [T_f(z)^T, T_{NN}(z)^T]^T$. Here the sufficient statistics $T(z)$ is defined by the concatenation of $T_f(z) = [T_1(z_1)^T, \cdots, T_d(z_d)^T]^T$ from a factorized exponential family and the outputs of a neural networks $T_{NN}(z)$ with universal approximation power. Then, the probability density can be written as:*

$$p_{\mathbf{T}, \lambda}(z|B, y) = \frac{\mathcal{Q}(z)}{\mathcal{C}(B, y)} \exp[T(z)^T \lambda(B, y)], \tag{2}$$

*where $\mathcal{Q}(z)$ is the base measure and $\mathcal{C}(B, y)$ is the normalizing constant.*

The above assumption is inspired by the recent advancement of iVAE [17]. However, there are two differences. Firstly, because Figure 1 allows for connections among the components $z_{ij}$ within $z$ as long as the generative model remains a valid DAG, our non-factorized conditional prior assumption is more general than the factorized distribution assumed by iVAE. In Equation 2, we utilize a neural networks $T_{NN}(z^c)$ for capturing the interactions among components of $z$. Secondly, our additionally observed auxiliary information is provided by multi-instance bag $B$ which may consist arbitrary number of instances instead of a single scalar or vector-valued instance $u$ as used by iVAE. Therefore, it is natural to ask (1) if identifiability still holds for non-factorized priors, and (2) how to aggregate information from bags of varying sizes for breaking the symmetry of the prior distribution?

To answer the above questions, we first describe our generative model and discuss its identifiability. Formally, let us consider the following generative models under Assumption 2:

$$p_\theta(x, z|B, y) = p_f(x|z)p_{\mathbf{T}, \lambda}(z|B, y), \tag{3}$$

$$p_f(x|z) = p_\varepsilon(x - f(z)), \tag{4}$$

where Equation 3 describes the generative process of $x$ given the underlying latent factors $z$, along with the bag context $B$ and the instance label $y$. Equation 4 implies that the observed representation $x$ is an additive noise function, i.e., $x = f(z) + \varepsilon$ where $\varepsilon$ is independent of $f$ or $z$.

Following standard VAE and iVAE derivation, the evidence lower bound for each instance $\boldsymbol{x}$ in bag $\boldsymbol{B}$ of the above generative model can be written as:

$$\mathcal{L}_{\mathrm{ELBO}} = \mathbb{E}_{q_{\boldsymbol{\phi}}(\boldsymbol{z}|\boldsymbol{x},\boldsymbol{B},y)}[\log p_{\boldsymbol{f}}(\boldsymbol{x}|\boldsymbol{z}) + \log p_{\boldsymbol{T},\boldsymbol{\lambda}}(\boldsymbol{z}|\boldsymbol{B},y) \tag{5}$$

$$- \log q_{\boldsymbol{\phi}}(\boldsymbol{z}|\boldsymbol{x},\boldsymbol{B},y)] + \log p(\boldsymbol{B}). \tag{6}$$

Note that the ELBO in Equation 6 contains an additional term of the bag prior $\log p(\boldsymbol{B})$ that do not affect identifiability but improves the estimation for the conditional prior [25]. Our main identifiability results can now be described as:

**Theorem 1.** *Assume we observe instances from multi-instance bags sampled according to Equation 2-4. Furthermore, assume that the following holds:*

 i. *The set $\{\boldsymbol{x} \in \mathcal{X} : \varphi_{\boldsymbol{\varepsilon}}(\boldsymbol{x} = 0\}$ has measure zero, where $\varphi_{\boldsymbol{\varepsilon}}$ is the characteristic function of the density $p_{\boldsymbol{\varepsilon}}$ defined in Equation 4.*

 ii. *The function $\boldsymbol{f}$ is injective and all of its second-order cross partial derivatives exist.*

 iii. *The sufficient statistics $T_{\boldsymbol{f}}$ are twice differentiable.*

 iv. *There exist $k + 1$ distinct bags $(\boldsymbol{B}^0, y^0), \ldots, (\boldsymbol{B}^d, y^d)$ such that the $k \times k$ matrix $E$ is invertible, where $k$ is the dimension of $\boldsymbol{T}$ and $E$ is defined as:*

$$E = (\boldsymbol{\lambda}(\boldsymbol{B}^1, y^1) - \boldsymbol{\lambda}(\boldsymbol{B}^0, y^0), \ldots, \boldsymbol{\lambda}(\boldsymbol{B}^k, y^k) - \boldsymbol{\lambda}(\boldsymbol{B}^0, y^0)) \tag{7}$$

*Then, the parameters $\boldsymbol{\theta} = (\boldsymbol{f}, \boldsymbol{T}, \boldsymbol{\lambda})$ are identifiable up to an equivalence class induced by permutation and componentwise transformations.*

It is interesting to observe that condition (iv) of Theorem 1 is easier to satisfy in MIL than standard supervised learning. Intuitively, the condition requires that the auxiliary information should "break the symmetry" in the representation space the model could learn. An intuitive analogy is inferring an object's shape from its shadow: if we only observe one shadow of the object, it's difficult to know its shape; however, if we observe multiple objects under similar lighting conditions (from a bag of instances), we may identify the lighting. If we observe an object under different lighting conditions (from many bags of instances), we may identify the underlying shape. Another analogy is to consider the bags as H&E stained histopathology images and the instances as image patches: in one bag, we observe cells under the same staining process, which provides information regarding the staining procedure; in different bags, we observe cancerous cells under different staining procedures, which makes it possible to infer the causal representations of cancerous cells. In standard supervised learning, instances are independent with no auxiliary information given. In MIL, instances within bags are naturally dependent and organized in bags, which may then provide the necessary auxiliary information for latent identifiability. We provide the proof for Theorem 1 in the Appendices.

Technically, condition (iv) can be restated as requiring the vectors $(\boldsymbol{\lambda}(\boldsymbol{B}^1, y^1) - \boldsymbol{\lambda}(\boldsymbol{B}^0, y^0), \ldots, \boldsymbol{\lambda}(\boldsymbol{B}^k, y^k) - \boldsymbol{\lambda}(\boldsymbol{B}^0, y^0))$ to be independent. Therefore, the instance label $y$ becomes unnecessary if there exist $k + 1$ distinct bags such that $E = (\boldsymbol{\lambda}(\boldsymbol{B}^1) - \boldsymbol{\lambda}(\boldsymbol{B}^0), \ldots, \boldsymbol{\lambda}(\boldsymbol{B}^k) - \boldsymbol{\lambda}(\boldsymbol{B}^0))$ of size $k \times k$ is invertible, which is especially attractive in MIL where the instance labels are not available but the bags are abundant.

Two practical issues need to be addressed before applying the identifiability theorems to solve MIL problems. The first hurdle lies in how to characterize the auxiliary information for conditioning the prior distribution of $\boldsymbol{z}$. This is different from standard iVAE where the auxiliary information is readily provided in a single-instance $\boldsymbol{u}$ since the auxiliary information in MIL are provided in the form of instances sampled from bags. To make things worse, the auxiliary information also cannot be derived from the bag labels or the multi-instance assumption. This is because they only tell us the bag labels and the relationship between instances labels and bag labels, instead of the generative process of the bags and their instances.

To solve the above problem, we propose to utilize a trainable permutation invariant function parameterized by deep neural networks. Formally, we define:

$$\boldsymbol{B}_i = net(\{\boldsymbol{x}_{i1}, \cdots, \boldsymbol{x}_{in_i}\}) = \rho[pool(\{\phi(\boldsymbol{x}_{i1}), \cdots, \phi(\boldsymbol{x}_{in_i})\})], \tag{8}$$

where $\rho$ and $\phi$ are arbitrary continuous functions implemented through neural networks, $pool$ is the sum operator over the instances $\boldsymbol{x}$ transformed by the neural network $\phi$, and finally $\rho$ is another neural

network that operates on the pooled transformations. An important property of the transformation defined in Equation 8 is that it is capable to express any permutation invariant function that operates on set inputs [39]. Since a multi-instance bags are both permutation invariant and contain instances of varying sizes [15], it is adequate to model its bag information as a set function of the instances.

The second missing piece of the puzzle is how to separate $z^c$ from $z^e$ in the identified latent factors $z$, since we would only require $z^c$ for the downstream tasks. To solve this problem, we take a different approach from a typical VAE-based disentanglement algorithm where their goal is to ensure the components of $z$, i.e., $z_p$ for $p = 1, \cdots, d$, are independent and correspond to meaningful semantics [13]. Although component-wise disentanglement is desirable for causal representation learning in general, due to the fact that multi-instance bags are only weakly supervised at the bag level, we argue that a more pragmatic approach is to utilize the bag labels for separating the causal factors $z^c$ of the instance labels from the non-causal ones $z^e$ without requiring additional information.

We now revisit the graphical model in Figure 1 to discuss the benefit of not pursuing component-wise disentanglement and using only $z^c$ for instance classification. The instance labels $y$ are correlated with both $z^c$ and $z^e$, and thus the entirety of $z$ is predictive of $y$. Therefore, for supervised learning tasks that assume i.i.d. training and test data, using $z$ is preferable for prediction. However, in MIL our goal is to predict the instance labels from unseen test bags. Since the bag-inherited factor $z^e$ may be different among training and test bags, using $z^e$ would negatively affect prediction. Furthermore, since instance classifiers in MIL must infer instance-level labels from bag-level weak supervision, excluding spurious correlations and including only the low dimensional causal representation also reduces the difficulty in learning the instance classifier. Another important benefit of separating $z^c$ from $z^e$ is that it also promotes OOD generalization. By considering bags as environments and the non-causal factors $z^e$ as the effects of environment-induced biases, using $z^c$ and excluding $z^e$ improves prediction robustness since $p(y|z^c)$ is invariant across bags as specified by Assumption 1(d), which also coincides with the goal of invariant causal prediction algorithms such as [4, 1, 23].

If the instance labels are known, then our goal of disentanglement can be straightforwardly achieved by utilizing the PC algorithm [33] to learn a Markov equivalent class of DAGs and find the direct causes as done in [23]. Unfortunately, this is not the case for MIL where only bag labels are known. To solve this problem, we propose to utilize the bag label $Y$ with the ELBO in Equation 6 using a bag-wise maximum operator. Specifically, we optimize for the following target

$$\mathcal{L}_{\text{CausalMIL}} = \log p_f(x^*|z^*) + \alpha \log p_\omega(Y|z^*) - \text{KL}[q_\phi(z^*|x^*, B)||p_{T,\lambda}(z^*|B)]$$
$$+ \log p_\vartheta(B|z) - \text{KL}[q_\psi(z|B)||p(B)], \tag{9}$$

where $z^* = \arg\max_B p_\omega(Y|z)$, and $p_\omega(Y|z)$ is a *linear* classifier with $\alpha$ as its weighting hyperparameter. It is also worth noting that the above ELBO is written with respect to a multi-instance bag instead of the instances. Furthermore, the last two terms can be obtained by deriving an ELBO for $\log p_\vartheta(B)$ in Equation 6 [25].

In other words, the reconstruction, classification, and conditional KL terms in the ELBO are optimized for *only one instance per bag*, and the instance is chosen using a maximum operator to satisfy the standard multi-instance assumption. The formulation of Equation 9 is effective for learning $z^c$ while ignoring $z^e$ because of the following two reasons. Firstly, as the ELBO is optimized per mini-batches consisted of many multi-instance bags using the reparameterization trick [18], the optimization process will force the linear classifier $p_\omega(y|z)$ to predict using the $z^c$ and ignore $z^e$; otherwise, the linearity of $p_\omega(y|z)$ will not be able to predict the instance labels since $z^e$ are different for different bags in the mini-batch. Moreover, because the reconstruction probabilities are evaluated at only one latent factor for each bag, the encoder $q_\phi(z|x, B)$ is encouraged to encode only $z^c$ since it carries the content causal information for reconstructing the positive $x$ and remains invariant across different bags in the mini-batch. To see this, consider the opposite scenario where the reconstruction losses are evaluated for every instance in the bag, the encoder would also encode $z^e$ since it remains the same within that bag. Therefore, CausalMIL is targeted to learn $z^c$ that are causal to the positive instances of a bag, and ignores both the non-causal factors $z^e$ and the causal factors for negative instances, as illustrated by the reconstructions in Figure 1 (right) and Figure 2.

It is worth noting that there exist two types of approximate inference in CausalMIL. The first one, common to almost all VAE-based algorithms, is the amortization gaps between the ELBOs and the marginal log-likelihoods caused by amortizing the variational parameters over the entire dataset, instead of optimizing for each training sample individually. The second one, specific to CausalMIL, is the approximation gap of only inferring the log-likelihood of one instances per each bag. However, as

Table 1: Results of instance label prediction performances on MNIST, FashionMNIST, and KuzushijiMNIST bags. The reported means $\pm$ std are the macro averaged over 10 one-vs-rest datasets, respectively.

| | MNIST Bags | | FashionMNIST Bags | | KuzushijiMNIST Bags | |
|---|---|---|---|---|---|---|
| | F-score | AUC-PR | F-score | AUC-PR | F-score | AUC-PR |
| mi-NET | 0.595±.078 | 0.702±.165 | 0.251±.290 | 0.329±.450 | 0.502±.106 | 0.616±.164 |
| Attn-MIL | 0.712±.147 | 0.776±.222 | 0.398±.243 | 0.534±.144 | 0.638±.145 | 0.654±.209 |
| KSA-MIL | 0.775±.092 | 0.845±.122 | 0.545±.288 | 0.617±.360 | 0.707±.084 | 0.782±.116 |
| MIVAE | 0.901±.035 | 0.921±.056 | 0.701±.257 | 0.733±.271 | 0.779±.147 | 0.838±.102 |
| CausalMIL | **0.966±.018** | **0.981±.018** | **0.748±.206** | **0.822±.168** | **0.833±0.109** | **0.934±.075** |

corroborated by our experiment results, these approximation behaviors does not prevent the success of CausalMIL in instance label prediction and OOD generalization.

# 4 Experiments

In this section, we first evaluate the instance label prediction performances of CausalMIL against MIL baselines including mi-Net [35], Attention-based MIL (Attn-MIL) [15], Kernel Self-Attention-based MIL (KSA-MIL) [29], Multi-Instance Variational Autoencoder (MIVAE) [40]. Then, we evaluate the out-of-distribution generalization ability of CausalMIL by comparing with supervised learning algorithms including ERM and its variants, Invariant Risk Minimization (IRM) [4], IRM with Game Theory (IRM GAME) [1], Invariant Causal Representation Learning (iCaRL) [23].

We implemented CausalMIL using PyTorch and conducted most of the experiments with a single NVIDIA RTX3090 GPU. Detailed settings, parameters, and data for reproducing the results are provided in the Appendices. The implementation code is publicly available at https://github.com/WeijiaZhang24/CausalMIL.

## 4.1 Instance Label Prediction

### 4.1.1 Datasets

Our quantitative evaluation first utilizes 30 multi-instance classification datasets generated from the MNIST [19], FashionMNIST [38], and KuzushijiMNIST [7] datasets. The bag construction procedure is similar to the multi-instance bags generation from the 20 Newsgroup corpus [43]. Specifically, we create the multi-instance MNIST-bags such that each bag contains a number of images where its bag size is drawn from a Gaussian distribution with fixed mean and variance. The bag is positive if it contains a target digit, e.g., '4', and negative if otherwise. Using 10 different digits as targets, we obtain 10 multi-instance MNIST-bag datasets. Similarly, we construct 10 FashionMNIST-bags and 10 KuzushijiMNIST-bags.

We also evaluate CausalMIL on a hematoxylin and eosin (H&E) stained Colon Cancer histopathology task [32]. Histopathology from whole-slide images is an important application of MIL since supervised predictions require pathologists to provide pixel-level annotations which is extremely expensive and time-consuming. The dataset contains 100 images obtained from tissues of either normal or malignant patient tissues. For each image bag, the instances are generated as patches $27 \times 27$ pixels using markings of major nuclei for each cell. A total amount of 22,444 instances (~220 instances per bag) are provided with ground truth instance labels, i.e. whether the cell is epithelial. A bag is labeled positive if it contains at least one epithelial patch and negative if otherwise.

### 4.1.2 Quantitative Results

For quantitative evaluation, we use the area under the Precision-Recall curve (AUC-PR), instead of area under the ROC curve due to class imbalance at the instance level. For example, although the original MNIST datasets are balanced, only ~5% of the instances are positive in the MNIST-bags. We also report the f-scores of the compared methods. For the methods that directly output instance prediction such as mi-NET, MIVAE, and CausalMIL, the results are obtained from thresholding their predicted scores. For methods based on the attention mechanism, such as Attn-MIL and KSA-MIL, the results are obtained by first normalizing the instance-to-bag attention weights to $[0, 1]$ and

Table 2: Instance label prediction performances of the Colon Cancer dataset. Experiments are repeated for 5 times and the average metric±standard deviation of 5-fold cross validations are reported.

| Method | F-score | AUC-PR |
|---|---|---|
| mi-Net | 0.392±.017 | 0.491±.028 |
| Attn-MIL | 0.466±.037 | 0.536±.014 |
| KSA-MIL | 0.510±.029 | 0.578±.025 |
| MIVAE | 0.675±.033 | 0.747±.032 |
| **CausalMIL** | **0.833±.024** | **0.878±.046** |

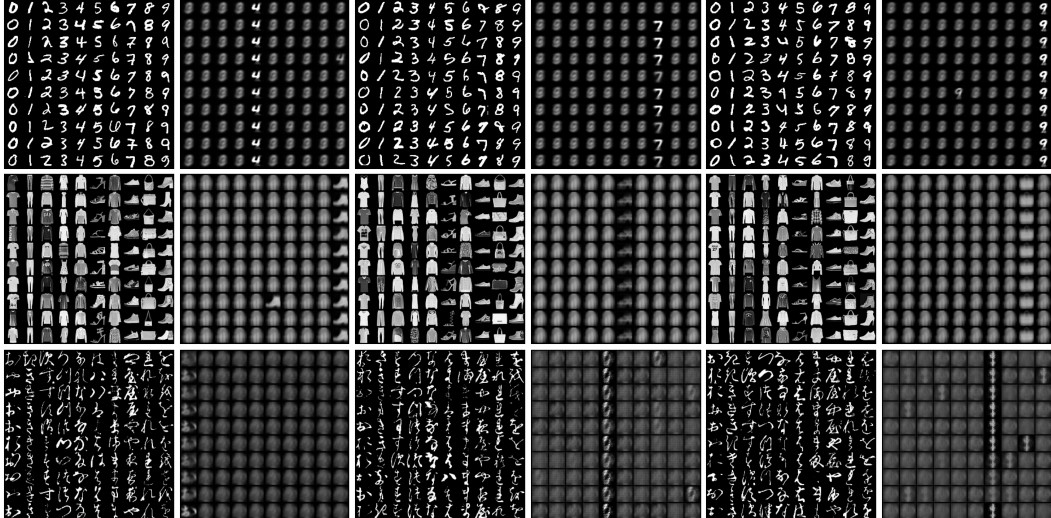

Figure 2: MNIST digits and CausalMIL reconstructions for digits '4', '7', '9'. Figures are best viewed when zoomed. FashionMNIST objects and CausalMIL reconstructions for 'dress', 'sandal', and 'handbag'. More results can be found in the Appendices. Figures best viewed when zoomed.

applying the threshold. As there is usually no instance label available in real application scenarios, we use $0.5$ uniformly as the classification threshold.

Table 1 shows the quantitative results on MNIST, FashionMNIST, and KuzushijiMNIST bags. CausalMIL performs significantly better than the compared methods on both metrics. Note that the large standard deviations for FashionMNIST-bags are because some fashion objects are much more difficult to classify (e.g., distinguishing shirts from pullovers and coats). Furthermore, we can see that attention-based MIL algorithms exhibit significant performance variations even for MNIST-bags. A possible reason is that when the data satisfy the standard multi-instance assumption, learning a weighted average of the individual instance contributions is not efficient. Among the compared algorithms, CausalMIL achieves the best performances with the lowest standard deviations.

Table 2 shows the instance prediction performances of the compared algorithms on the Colon Cancer dataset. The result trends are similar to previous ones. For discriminative deep learning-based algorithms, attention-pooling (Attn-MIL) performs better than max-pooling (mi-Net), while incorporating the self-attention mechanism (KSA-MIL) further improves the performances. A recent VAE-based generative MIL algorithm (MIVAE) performs better than the discriminative algorithms but still significantly worse than CausalMIL because it does not learn meaningful representations and does not provide identifiability guarantee.

### 4.1.3 Qualitative Results

We now qualitatively evaluate the inferred latent factors of CausalMIL by examining the reconstructions of decoder $p_f(x|z)$ using the inferred latent factors in Figure 2. For example, in the first row we show the original images and reconstructions of MNIST-bags using '4', '7', and '9' as positive instances. In the reconstructions of '4' we can see that CausalMIL learns the semantic causal characteristics of the positive instances while ignoring the others. The '4's are reconstructed with a

Table 3: Classification performances on distributionally biased ColoredMNIST and ColoredFashionMNIST datasets in terms of accuracy ± standard deviation. CausalMIL (ours) is weakly supervised whereas the other approaches are supervised.

| | ColoredMNIST | | ColoredFashionMNIST | |
|---|---|---|---|---|
| | Train | Test | Train | Test |
| ERM | 0.849±.002 | 0.105±.007 | 0.832±.010 | 0.225±.007 |
| ERM1 | 0.848±.002 | 0.109±.005 | 0.813±.014 | 0.333±.089 |
| ERM2 | 0.850±.002 | 0.101±.002 | 0.844±.019 | 0.132±.008 |
| ROBUST MIN MAX | 0.843±.004 | 0.152±.025 | 0.828±.001 | 0.292±.086 |
| IRM | 0.593±.044 | 0.628±.096 | 0.750±.003 | 0.553±.124 |
| IRM GAME | 0.634±.011 | 0.599±.027 | 0.690±.101 | 0.702±.015 |
| iCaRL | 0.706±.008 | 0.688±.007 | 0.750±.004 | 0.736±.006 |
| MIVAE | 0.810±.005 | 0.156±.003 | 0.823±.003 | 0.284±.120 |
| **CausalMIL** | **0.919±.002** | **0.892±.002** | **0.873±.007** | **0.816±.015** |

'U' shape at the top and a short line at the bottom, while other digits are reconstructed to noise. The 'U' shape of '4' is clear because it is crucial for the positive instance; however, the line at the bottom is blurry since it can be written in different angles and the angles are irrelevant. The reconstructions from FashionMNIST-bags and KuzushijiMNIST-bags datasets also corroborate the above results.

## 4.2 Out-of-Distribution Generalization

To further validate whether CausalMIL identifies the causal representations $z^c$, we evaluate CausalMIL on out-of-distribution (OOD) generalization tasks and compare it supervised algorithms designed specifically for OOD generalization [4, 1, 23].

We utilize two widely-adopted biased classification tasks, ColoredMNIST and ColoredFashionMNIST that are commonly used in the invariant risk minimization literature [4]. The datasets are constructed following the same procedure as in IRM, where the task is to classify whether a digit is less than 5. The instance labels are first binarized and randomly flipped with 25% probability. Then, the images are colored by green with probabilities $p_e$ which vary across environments, while the rest $1 - p_e$ of the images are colored by red. There are three environments (two training, one test) where $p_e = 0.1, 0.2$, and $0.9$, respectively. Therefore, simply predicting colors instead of digits will lead to high accuracy in the training environments, but the correlation is reversed in the test environment. For CausalMIL, we construct multi-instance bags based on the flipped instance labels utilizing instances from both training environments. Only the bag labels are provided to CausalMIL during training, while the instances from the test environment are used for testing.

We compare the instance label prediction performances of CausalMIL against two categories of *supervised algorithms*. The first category includes standard Empirical Risk Minimization using CNN trained with both (ERM) and each of the training environment (ERM1 and ERM2), and ROBUST MIN MAX which minimizes the maximum loss across different environments. The second category are specifically designed OOD generalization algorithms including Invariant Risk Minimization(IRM) [4], IRM GAME [1], and iCaRL [23]. Considering that most of the baseline results come from iCaRL [23], we set CausalMIL to use the same convolutional structure as [23] for fair comparison.

The results are reported in Table 3. We can see that CausalMIL performs significantly better than all compared supervised methods. There are two reasons behind this. The first reason is that CausalMIL effectively forces the biases into $z^e$ by disentangling the causal from non-causal factors, and uses only $z^c$ for prediction. Although previous supervised OOD generalization algorithms, i.e., IRM and iCaRL, aim to identify and only use the causes of $y$ to improve robustness, they mainly rely on the instance-level labels for identifiability and utilize the scarcely available environments for robustness. Unfortunately, as the instance-level labels are biased from the training environments, their approaches are less effective. On the contrary, in CausalMIL we rely on the abundance of bag information for identification. The second reason is that the formulation of MIL is naturally more robust to instance-level label noises as positive bags naturally consist of both positive and negative instances. Furthermore, previous theoretical results have also shown that generative MIL algorithm [9] is effective even when negative bags are falsely labeled as positive.

# 5   Related Work

Most MIL algorithms can be categorized into two groups according to whether they work at the bag space or at the instance space. Bag space algorithms [43, 37, 41, 10] work by embedding multi-instance bags into a single feature vector representation and then solving the single-instance learning problem in the embedded space, and therefore are capable of predicting instance labels. Instance space MIL algorithms aim to directly separate the positive instances from the negative ones [2, 16, 12, 34]. These algorithms can be used to predict instance labels, although not all of them are explicitly designed for the task.However, these algorithms only work with pre-computed features.

Recently, several deep learning-based MIL algorithms have been proposed by utilizing permutation-invariant pooling operations, and can be used for instance label prediction. For example, [35] used a max-pooling layer with a fully connected neural network; [15] introduced the attention mechanism as a permutation-invariant MIL pooling operation and used the attention weights for instance label prediction; [29] proposed to integrate self-attention with the attention mechanism for capturing the non-i.i.d. information among instances; [21] utilized Gumbel reparametrization and proposed an algorithm for the generalized MIL assumption. Unfortunately, none of the above methods learn semantically meaningful representations.

Unsupervised learning of meaningful representations with VAE-based models has attracted much attention [13, 6]. Since it has been shown that unsupervised disentanglement is theoretically impossible without inductive biases [22], various methods have been proposed for learning disentangled representations using different forms of weak supervisions. One form of weakly-supervised disentanglement learning that is closely related to MIL is group-based VAE [5, 14]. Similar to MIL where instances are organized into bags, group-based algorithms require the objects to be divided into groups: those within the same group share the same *content* but have different *styles* that are independent of the contents, while those among different groups should have different contents.

Recently, MIVAE [40] is proposed based on connections between MIL and group-based disentangled representation learning which utilizes VAE for capturing the dependencies among instances as the shared content factors. However, MIVAE also does not learn semantically meaningful latents, nor does it provide identifiability on what latents it actually learns because of its unconditional prior. Please refer to the Appendices for a more detailed comparison between CausalMIL and MIVE.

# 6   Conclusion

In this work, we proposed CausalMIL, an algorithm that learns semantically meaningful causal representations from multi-instance bags and the corresponding bag-level supervision. The learned representations not only significantly improve the performances of traditional MIL tasks such as instance label prediction, but also exhibits notable performances in supervised learning tasks such as OOD generalization. Qualitative and quantative evaluation results show that CausalMIL performs significantly better than existing state-of-the-art deep MIL algorithms on semi-synthetic and real-world datasets.

There are some limitations of our method and many future directions worth investigating. For example, CausalMIL is only designed for the standard multi-instance assumption. It would be interesting to see if the results extend to other assumptions in multi-instance learning. A possible way for such extension is to utilize the Gumbel reparameterization as discussed in [21]. Another possible direction would be developing an extension to learning from positive unlabeled (PU) examples, another weakly supervised learning problem that is closely related to MIL.

## Acknowledgments and Disclosure of Funding

The authors wish to thank the anonymous reviewers for their constructive comments and suggestions. This work was supported by the National Science Foundation of China (62176055, 62206047, 72204110). We thank the Big Data Center of Southeast University for providing the facility support on the numerical calculations in this paper.

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
