# A  Appendices

## A.1  Additional Experiment Results

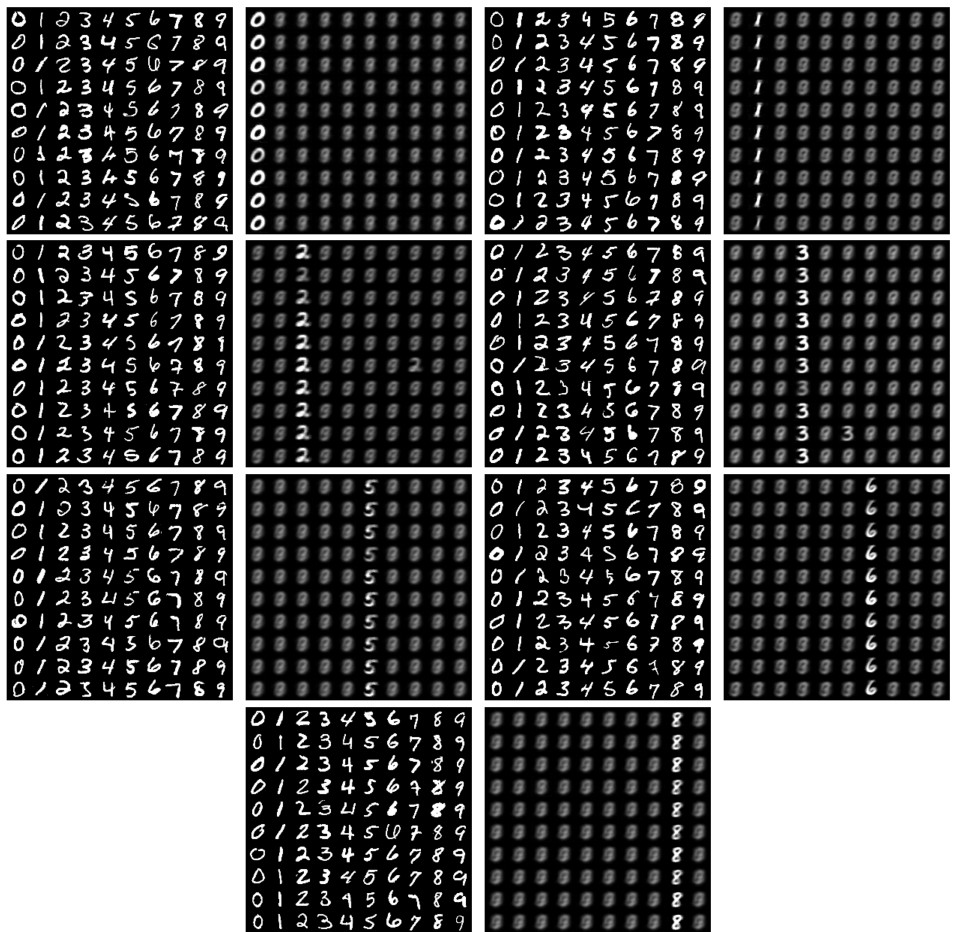

Figure 1: (a) Original MNIST digits and CausalMIL reconstructions for digits '0', '1', '2', '3', '5', '6', and '8'. The original digits are randomly sampled from the test set. Figures are best viewed when zoomed.

We now report the qualitative results on MNIST, FashionMNIST, and KuzushijiMNIST datasets that are omitted from the main manuscript. We can see that on MNIST-bags, CausalMIL successfully learns semantically meaningful representations for all the digits. On FashionMNIST-bags, TargetedMI makes some mistakes among the 'pullover', 'coat', and 'shirt' objects (the 3rd, 5th, and 7th columns). This is perhaps because that these objects themselves are difficult to distinguish even for humans, and the original FashionMNIST labels are not noise free. For example, looking at the subfigure at the 1st row, 1st column, we can see that some objects belonging to the 'shirt' class are not different from those in the 't-shirt' class (for example, the object at the 3rd row, 7th column). Furthermore, it is also difficult to differentiate between 'pullover' and 'shirt' (the 3rd and the 7th column of each subfigure).

An interesting observation from the KuzushijiMNIST-bags is that CausalMIL is able to learn the causal invariant representation for hiragana characters that have more than one handwritten forms. Let us look at the bottom subfigure which depicts 10 hiragana characters and their handwriting. We can see that some of the hiragana character have two types of handwritten forms. For example, the second hiragana character has two forms from the bottom subfigure; accordingly, in its original vs reconstructions comparisons (first two subfigures of the first row), CausalMIL is successful in recognizing these two different handwritten forms.

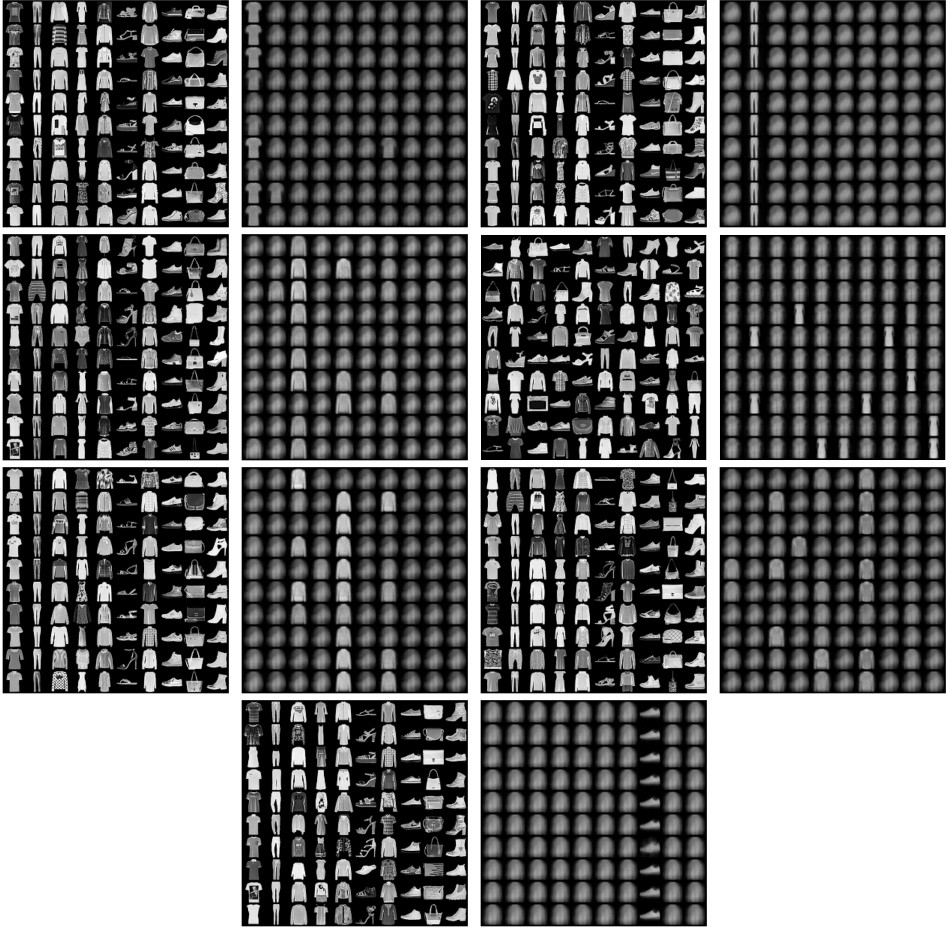

Figure 2: (a) Original FashionMNIST images and CausalMIL reconstructions for object classes 't-shirt', 'trousers', 'pullover', 'coat', 'shirt', 'sneaker', and 'ankle boot'. The original images are randomly sampled from the test set. Figures are best viewed when zoomed.

## A.2 Ablation Results

In this section we conduct ablation studies of CausalMIL. Specifically, in Figure 4 we show (1) the reconstructions of a standard Autoencoder (AE); (2) the reconstructions of CausalMIL without the KL regularization term (miAE, a multi-instance Autoencoder with max-pooling); (3) the reconstructions of a non-targeted version of CausalMIL (miVAE a multi-instance Variational Autoencoder and we denote it as miVAE); and (4) the reconstructions of MIVAE [4] (another multi-instance Variational Autoencoder that learns an instance-specific latent factor and a bag-level shared latent factor similar to group-based VAE methods).

From the reconstruction results of AE, miAE and miVAE, we can see that both targeted reconstruction and the bag-dependent KL divergence is necessary for learning meaningful representations: without the bag conditional KL term, the representations learned by miAE is similar to a vanilla Autoencoder. Without the targeted reconstruction, the latent representation learned by miVAE is indistinguishable to human eyes.

From the reconstructions of MIVAE (which is trained on bags with '9' versus those without), we can see that it does learn the representation of '9'. However, the representations are very noisy: firstly, it also learns the representations of other digits, such as '0' and '1', and many other digits have been incorrectly reconstructed to '0' and '1'. Secondly, it reconstructs digits that do not resemble '9', such as the '6' at the 1st row into '9', 7th column. Theses noises in the latents are caused by the fact that MIVAE does not have identifiability guarantee.

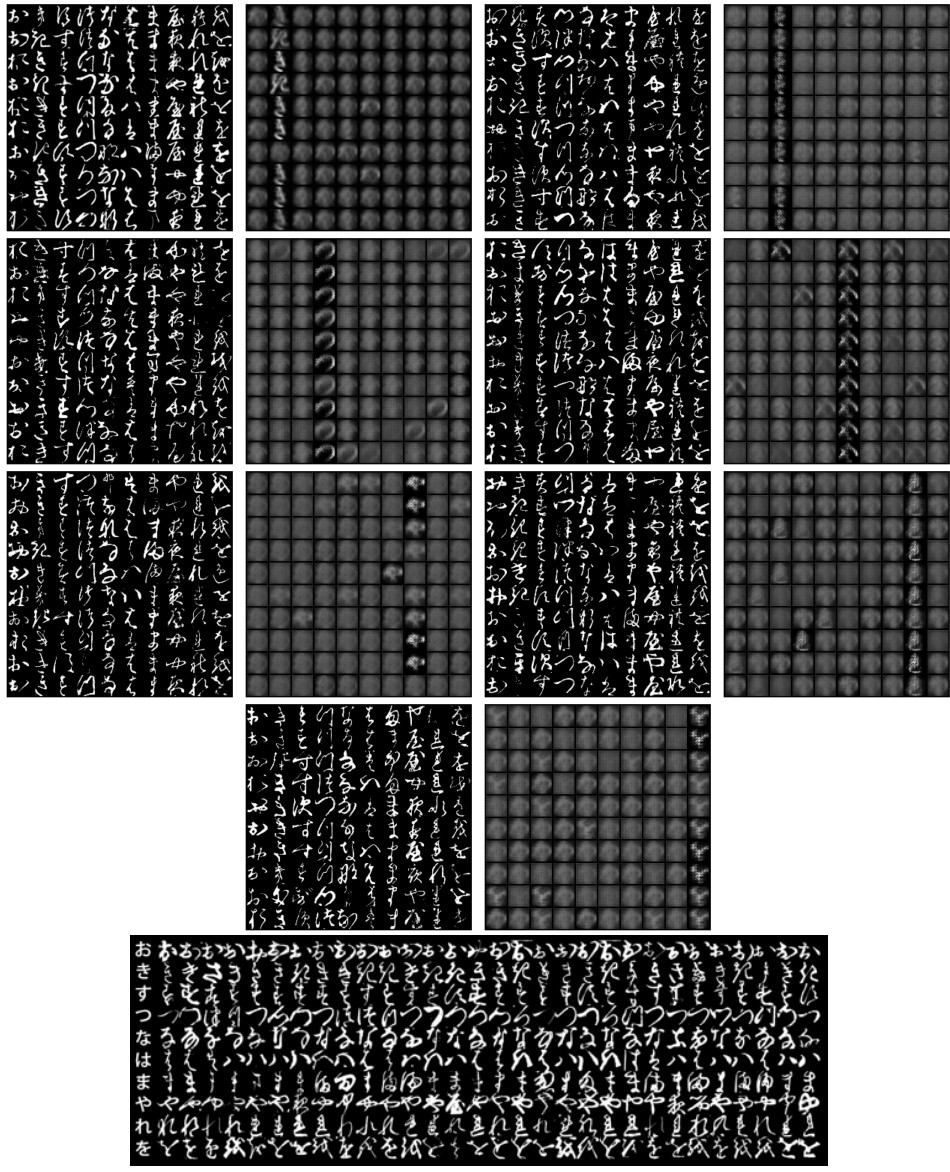

Figure 3: (a) Original KMNIST characters and CausalMIL reconstructions. The original digits are randomly sampled from the test set. The figure in the last row contains original Kuzushiji characters for all 10 classes: the first column of this subfigure shows each character in its modern hiragana form, while the other columns are each character's handwritten forms. These figures are best viewed when zoomed.

## A.3 Sensitivity to Bag Sizes and Witness Rates

Figure 5 shows how different bag sizes and witness rates (number of positive instances per bag) affect the instance prediction performances of different MIL algorithms. From the bag size results, we can see that the performances of all compared MIL algorithms decrease as the bag size increases, which is not surprising as it affects the levels of inexactness in the supervision; however, the performances drop of CausalMIL is significantly less severe than the compared ones. From the witness rate results, it is interesting to observe that larger witness sometimes decreases performances for algorithms that consider instances as independent (e.g., mi-Net and Attn-MIL). Again, CausalMIL performs consistently better than the compared algorithms.

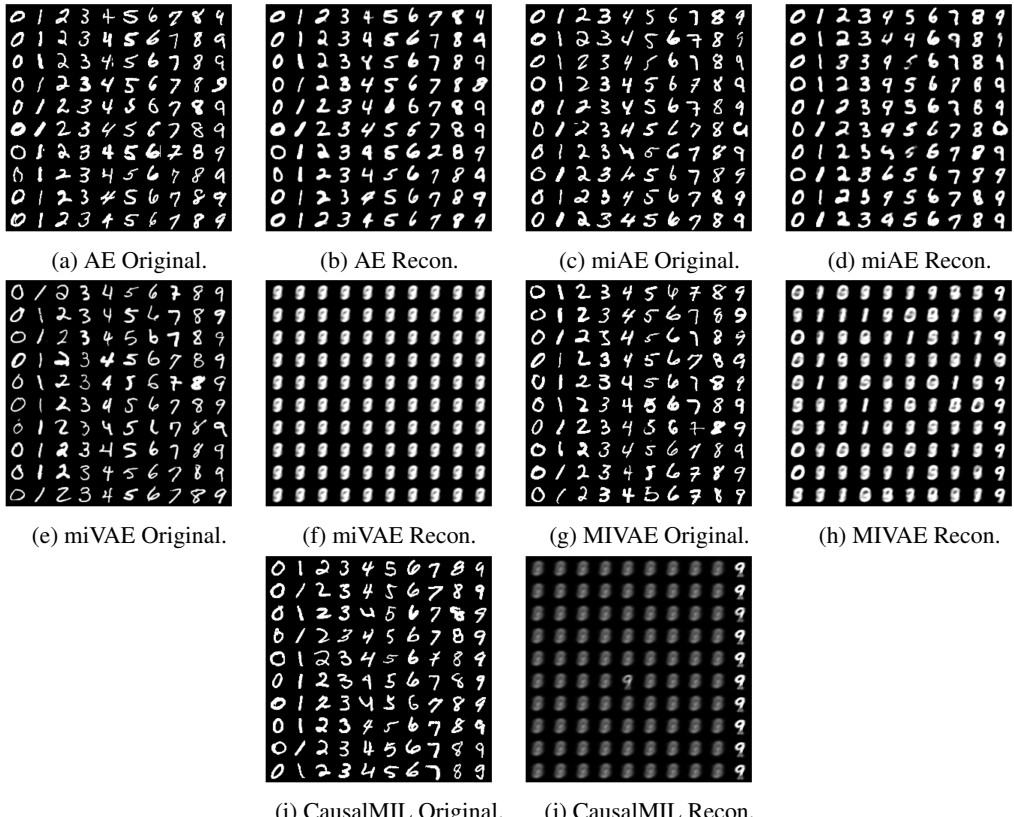

(a) AE Original.  (b) AE Recon.  (c) miAE Original.  (d) miAE Recon.

(e) miVAE Original.  (f) miVAE Recon.  (g) MIVAE Original.  (h) MIVAE Recon.

(i) CausalMIL Original.  (j) CausalMIL Recon.

Figure 4: (a) Original digits and reconstructions learned for the digit '9' of ablations and CausalMIL. These figures are best viewed when zoomed.

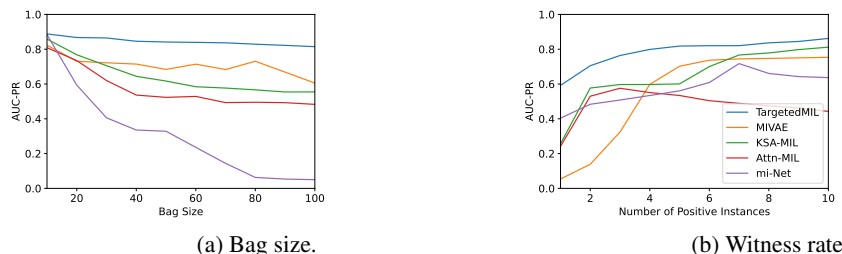

(a) Bag size.  (b) Witness rate.

Figure 5: The average AUC-PR of different bag sizes (left) and witness rates (right) on ten FashionMNIST-bags.

## B  Proof of Identifiability

In this section, we provide the proof for the latent identifiability of the model in Figure 1. Our proof consists of three main components. It is worth noting that compared to the proof of iVAE [1], the important differences in our proof are located in the third step.

In the first step, we use the first assumption in Theorem 1 to demonstrate that the observed data distributions are equivalent to the noiseless distributions. Specifically, suppose that we have two sets of parameters $(\boldsymbol{f}, \boldsymbol{T}, \boldsymbol{\lambda})$ and $(\tilde{\boldsymbol{f}}, \tilde{\boldsymbol{T}}, \tilde{\boldsymbol{\lambda}})$, with a change of variable $\overline{\boldsymbol{x}} = \boldsymbol{f}(\boldsymbol{z}) = \tilde{\boldsymbol{f}}(\boldsymbol{z})$, we have:

$$\tilde{p}_{\boldsymbol{T},\boldsymbol{\lambda},\boldsymbol{f},\boldsymbol{B},y}(\boldsymbol{x}) = \tilde{p}_{\tilde{\boldsymbol{T}},\tilde{\boldsymbol{f}},\tilde{\boldsymbol{\lambda}},\boldsymbol{B},y}(\boldsymbol{x}). \tag{9}$$

*Proof.* For simplicity of notations we denote $(\boldsymbol{B}, y)$ by $\boldsymbol{U}$, we have

$$p_{\boldsymbol{\theta}}(\boldsymbol{x}|\boldsymbol{U}) = p_{\tilde{\boldsymbol{\theta}}}(\boldsymbol{x}|\boldsymbol{U}) \tag{10}$$

$$\implies \int p_\epsilon(\boldsymbol{x} - \boldsymbol{f}(\boldsymbol{z}))p_{\boldsymbol{T},\boldsymbol{\lambda}}(\boldsymbol{z}|\boldsymbol{U})d\boldsymbol{z} = \int p_\epsilon(\boldsymbol{x} - \tilde{\boldsymbol{f}}(\boldsymbol{z}))p_{\tilde{\boldsymbol{T}},\tilde{\boldsymbol{\lambda}}}(\boldsymbol{z}|\boldsymbol{U})d\boldsymbol{z} \tag{11}$$

$$\implies \int p_\epsilon(\boldsymbol{x} - \bar{\boldsymbol{x}})p_{\boldsymbol{T},\boldsymbol{\lambda}}(\boldsymbol{f}^{-1}(\bar{\boldsymbol{x}}|\boldsymbol{U})vol(J_{f^{-1}}(\bar{\boldsymbol{x}}))d\bar{\boldsymbol{x}} = \int p_\epsilon(\boldsymbol{x} - \bar{\boldsymbol{x}})p_{\boldsymbol{T},\boldsymbol{\lambda}}(\tilde{\boldsymbol{f}}^{-1}(\bar{\boldsymbol{x}}|\boldsymbol{U})vol(J_{\tilde{f}^{-1}}(\bar{\boldsymbol{x}}))d\tilde{\boldsymbol{x}}$$
$$\tag{12}$$

$$\implies \int p_\epsilon(\boldsymbol{x} - \bar{\boldsymbol{x}})\tilde{p}_{\boldsymbol{T},\boldsymbol{\lambda},\boldsymbol{f},\boldsymbol{U}}(\bar{\boldsymbol{x}})d\bar{\boldsymbol{x}} = \implies \int p_\epsilon(\boldsymbol{x} - \bar{\boldsymbol{x}})\tilde{p}_{\tilde{\boldsymbol{T}},\tilde{\boldsymbol{f}},\tilde{\boldsymbol{\lambda}},\boldsymbol{U}}(\bar{\boldsymbol{x}})d\bar{\boldsymbol{x}}' \tag{13}$$

$$\implies (\tilde{p}_{\boldsymbol{T},\boldsymbol{\lambda},\boldsymbol{f},\boldsymbol{U}} * p_\varepsilon)(\boldsymbol{x}) = (\tilde{p}_{\tilde{\boldsymbol{T}},\tilde{\boldsymbol{f}},\tilde{\boldsymbol{\lambda}},\boldsymbol{U}} * p_\varepsilon)(\boldsymbol{x}) \tag{14}$$

$$\implies F[\tilde{p}_{\boldsymbol{T},\boldsymbol{\lambda},\boldsymbol{f},\boldsymbol{U}}](\boldsymbol{\omega})\phi_\varepsilon(\boldsymbol{\omega}) = F[\tilde{p}_{\tilde{\boldsymbol{T}},\tilde{\boldsymbol{f}},\tilde{\boldsymbol{\lambda}}}](\boldsymbol{\omega})\phi_\varepsilon(\boldsymbol{\omega}) \tag{15}$$

$$\implies F[\tilde{p}_{\boldsymbol{T},\boldsymbol{\lambda},\boldsymbol{f},\boldsymbol{U}}](\boldsymbol{\omega}) = F[\tilde{p}_{\tilde{\boldsymbol{T}},\tilde{\boldsymbol{f}},\tilde{\boldsymbol{\lambda}},\boldsymbol{U}}](\boldsymbol{\omega}) \tag{16}$$

$$\implies \tilde{p}_{\boldsymbol{T},\boldsymbol{\lambda},\boldsymbol{f},\boldsymbol{U}}(\boldsymbol{x}) = \tilde{p}_{\tilde{\boldsymbol{T}},\tilde{\boldsymbol{f}},\tilde{\boldsymbol{\lambda}},\boldsymbol{U}}(\boldsymbol{x}). \tag{17}$$

where

- in Equation 12, $J$ denotes the Jacobian, $vol(B) = \sqrt{\det(B^T B)}$.

- in Equation 13, we introduced

$$\tilde{p}_{\boldsymbol{T},\boldsymbol{\lambda},\boldsymbol{f},\boldsymbol{U}} \triangleq p_{\boldsymbol{T},\boldsymbol{\lambda}}(\boldsymbol{f}^{-1})(\boldsymbol{x}|\boldsymbol{U})vol(J_{f^{-1}}(\boldsymbol{x}))\mathbb{I}(\boldsymbol{x}) \tag{18}$$

- in Equation 14, $*$ denotes the convolution operator.

- in Equation 15, $F$ denotes the Fourier transformation and $\phi_\varepsilon = F[p_\varepsilon]$.

- in Equation 16, $\phi_\varepsilon(\boldsymbol{w})$ is dropped because it is non-zero almost everywhere according to the first assumption of Theorem 1.

$\square$

In the second step, the fourth assumption in Theorem 1 is used for removing all the terms that are a function of $\boldsymbol{x}$ or $\boldsymbol{B}$. By substituting $p_{\boldsymbol{T},\boldsymbol{\lambda}}$ with its exponential conditionally factorial form, we can show that:

$$\boldsymbol{T}(\boldsymbol{f}^{-1})(\boldsymbol{x}) = \boldsymbol{A}\tilde{\boldsymbol{T}}(\tilde{\boldsymbol{f}}^{-1})(\boldsymbol{x}) + \boldsymbol{c} \tag{19}$$

*Proof.* Using Equation 18 to substitute Eauation 17, we have

$$p_{\boldsymbol{T},\boldsymbol{\lambda}}(\boldsymbol{f}^{-1})(\boldsymbol{x}|\boldsymbol{U})vol(J_{f^{-1}}(\boldsymbol{x}))\mathbb{I}(\boldsymbol{x}) = p_{\tilde{\boldsymbol{T}}},\tilde{\boldsymbol{\lambda}}(\tilde{\boldsymbol{f}}^{-1})(\boldsymbol{x}|\boldsymbol{U})vol(J_{\tilde{f}^{-1}}(\boldsymbol{x}))\mathbb{I}(\boldsymbol{x}). \tag{20}$$

Then, we can apply logarithm on the above equation and substitute $p_{\boldsymbol{T},\boldsymbol{\lambda}}$ with its definition in Equation 3, and obtain

$$\log vol(J_{f^{-1}}(\boldsymbol{x})) \log Q(\boldsymbol{f}^{-1}\boldsymbol{x}) - \log Z(\boldsymbol{U}) + \langle \boldsymbol{T}(\boldsymbol{f}^{-1}(\boldsymbol{x})), \boldsymbol{\lambda}(\boldsymbol{U}) \rangle \tag{21}$$
$$= \log vol(J_{\tilde{f}^{-1}}(\boldsymbol{x})) \log \tilde{Q}(\tilde{\boldsymbol{f}}^{-1}\boldsymbol{x}) - \log \tilde{Z}(\boldsymbol{U}) + \langle \tilde{\boldsymbol{T}}(\tilde{\boldsymbol{f}}^{-1}(\boldsymbol{x})), \tilde{\boldsymbol{\lambda}}(\boldsymbol{U}) \rangle \tag{22}$$

Let $\boldsymbol{U}^0, \cdots, \boldsymbol{U}^k$ be the $k+1$ points defined in the fourth assumption of Theorem 1, we can obtain $k+1$ equation. By subtracting the first equation from the remaining $k$ equations, we then obtain:

$$\langle \boldsymbol{T}(\boldsymbol{f}^{-1}(\boldsymbol{x})), \boldsymbol{\lambda}(\boldsymbol{U}^l) - \boldsymbol{\lambda}(\boldsymbol{U}^0) \rangle + \log \frac{Z(\boldsymbol{U}^0)}{Z(\boldsymbol{U}^l)}$$
$$= \langle \tilde{\boldsymbol{T}}(\tilde{\boldsymbol{f}}^{-1}(\boldsymbol{x})), \tilde{\boldsymbol{\lambda}}(\boldsymbol{U}^l) - \tilde{\boldsymbol{\lambda}}(\boldsymbol{U}^0) \rangle + \log \frac{\tilde{Z}(\boldsymbol{U}^0)}{\tilde{Z}(\boldsymbol{U}^l)}, \tag{23}$$

where $l = 1, \cdots, k$. Let $\boldsymbol{b} \in \mathbb{R}^k$ in which $b_l = \log \frac{\tilde{Z}(\boldsymbol{U}^0)Z(\boldsymbol{U}^l)}{\tilde{Z}(\boldsymbol{U}^l)Z(\boldsymbol{U}^0)}$, we have

$$L^T \boldsymbol{T}(\boldsymbol{f}^{-1}(\boldsymbol{x})) = \tilde{L}\tilde{\boldsymbol{T}}(\tilde{\boldsymbol{f}}^{-1}(\boldsymbol{x})) + \boldsymbol{b}. \tag{24}$$

Finally, we multiply both side by $L^{-T}$ and obtain

$$\boldsymbol{T}(\boldsymbol{f}^{-1}(\boldsymbol{x})) = A\tilde{\boldsymbol{T}}(\tilde{\boldsymbol{f}}^{-1}(\boldsymbol{x})) + \boldsymbol{c}, \tag{25}$$

where $A = L^{-T}L$ and $\boldsymbol{c} = L^{-T}\boldsymbol{b}$. $\qquad\square$

In the third step, we show that the transformation $A$ is invertible, such that:

$$(\boldsymbol{f}, \boldsymbol{T}, \boldsymbol{\lambda}) \sim (\tilde{\boldsymbol{f}}, \tilde{\boldsymbol{T}}, \tilde{\boldsymbol{\lambda}}), \tag{26}$$

*Proof.* We start by evaluating Equation 25 at $k+1$ points of $\boldsymbol{z}_l, \boldsymbol{x}_l$ and obtain $k+1$ equations. Then, we subtract the first equation from the remaining $k+1$ equations:

$$[\boldsymbol{T}(\boldsymbol{z}_1) - \boldsymbol{T}(\boldsymbol{z}_0), \cdots, \boldsymbol{T}(\boldsymbol{z}_k) - \boldsymbol{T}(\boldsymbol{z}_0)]$$
$$= A[\tilde{\boldsymbol{T}}(\tilde{\boldsymbol{f}}^{-1}(\boldsymbol{x}_1)) - \tilde{\boldsymbol{T}}(\tilde{\boldsymbol{f}}^{-1}(\boldsymbol{x}_0)), \cdots, \tilde{\boldsymbol{T}}(\tilde{\boldsymbol{f}}^{-1}(\boldsymbol{x}_l)) - \tilde{\boldsymbol{T}}(\tilde{\boldsymbol{f}}^{-1}(\boldsymbol{x}_0))]. \tag{27}$$

Next we only need to show that for $\boldsymbol{z}_0$ there exist $k$ points $\boldsymbol{z}_1, \cdots, \boldsymbol{z}_k$ such that the columns are linear independent, which can be proven by contradiction. Suppose that there exists no such $\boldsymbol{z}_1, \cdots, \boldsymbol{z}_k$, then $\langle \boldsymbol{T}(\boldsymbol{z}) - \boldsymbol{T}(\boldsymbol{z}_0), \boldsymbol{\lambda} \rangle = 0$ and thus $\boldsymbol{T}(\boldsymbol{z}) = \boldsymbol{T}(\boldsymbol{z}_0) = \text{const}$. This contradicts with the assumption that the prior distribution is strongly exponential. Therefore, there must exist $k+1$ points such that the transformation is invertible. $\qquad\square$

## C  Implementation and Experiment Details

### C.1  Details and Parameters

CausalMIL is implemented using PyTorch 1.12. In the implementation, the term $\log p_{\vartheta}(\boldsymbol{B}|\boldsymbol{z})$ is omitted from the ELBO as $\boldsymbol{z}^c$ is independent of $\boldsymbol{B}$. The unconditional priors are assumed to be i.i.d sampled from $\mathcal{N}(\boldsymbol{0}, \boldsymbol{1})$. Furthermore, we enforce the bag transformation mapping $\phi$ to share the same parameters as those of the encoder.

For all experiments, the CausalMIL models are trained with AdamW optimizer for 200 epochs, and the best epochs are selected simply according to the training ELBO. The parameters and the best epoch of CausalMIL are also tuned using the training ELBO. To show that our results are indeed identifiable instead of cherry picking the reconstructions, for all of the MNIST-bags, FashionMNIST-bags, KuzushijiMNIST-bags experiments the parameters are fixed to the same value tuned according to MNIST-bags, i.e., we set learning rate to $1e-3$, weight for max-pooling to 1000, weight for the KL divergences to 1. The latent dimensions for $\boldsymbol{z}$ is set to 24 for all datasets. The optimization methods and parameters for the compared methods selected according to the strategy provided in their publicly available implementations and in their publications. We also performed extensive grid search for parameters of the compared algorithms. however, we find that this does not significantly improve their performances.

### C.2  Network structures

For the MNIST, FashionMNIST, and KuzushijiMNIST bags, we use the convolutional encoder and decoder illustrated in Table 1.

Table 1: Encoding and decoding network structures for MNIST, FashionMNIST and KuzushijiMNIST bags.

| Layer | Type |
|---|---|
| 1 | conv2d(4,2,1)-32 + BatchNorm + ReLU |
| 2 | conv2d(4,2,1)-128 + BatchNorm + ReLU |
| 3 | conv2d(7,1,0)-512 + BatchNorm + ReLU |

| Layer | Type |
|---|---|
| 1 | ConvTranspose2d(7,1,0)-512 + BatchNorm +ReLU |
| 2 | ConvTranspose2d(4,2,1)-128 + BatchNorm + ReLU |
| 3 | ConvTranspose2d(4,2,1)-32 + BatchNorm + ReLU |

For the Colon Cancer histopathology dataset, we use the same convoluntional network structures as the compared methods for fairness of comparison. For encoding the images, we used the following

convolutional network (Table 2 (left)) as used in [1]. For decoding the from the latent factors, we use a corresponding de-convolutional network as specified in Table 2 (right).

Table 2: Network structures for the Colon Cancer dataset.

| Layer | Type | Layer | Type |
|---|---|---|---|
| 1 | conv(4,1,0)-36 + ReLU | 1 | Upsampling(10) |
| 2 | maxpool(2,2) | 2 | ConvTranspose(3,1,0)-48 +ReLU |
| 3 | conv(3,1,0)-48 + ReLU | 3 | Upsampling(5) |
| 4 | maxpool(2,2) | 4 | ConvTranspose(4,1,0)-36 +ReLU |

Furthermore, for the biased datasets we use the same network structures as in the causal invariant representation learning literature [4, 1, 2] as illustrated in Table 3.

Table 3: Encoding and decoding network structures for the distribution biased datasets [4].

| Layer | Type |
|---|---|
| 1 | conv2d(3,2,1)-32 + ReLU |
| 2 | conv2d(3,2,1)-32 + ReLU |
| 3 | conv2d(3,1,0)-32 + ReLU |

| Layer | Type |
|---|---|
| 1 | ConvTranspose2d(3,2, 1)-32 +ReLU |
| 2 | ConvTranspose2d(3,2,1)-32 + ReLU |
| 3 | ConvTranspose2d(3,2,1)-3 + ReLU |

## C.3 Data and Code Availability

Pre-processed multi-instance bags for the Colon Cancer dataset can also be downloaded directly from https://drive.google.com/file/d/1RcNlwg0TwaZoaFO0uMXHFtAo_DCVPE6z/view?usp=sharing. The original images of this dataset are available at https://warwick.ac.uk/fac/sci/dcs/research/tia/data/crchistolabelednucleihe/.

For the compared methods, their implementations can be found at:
mi-Net is implemented at https://github.com/yanyongluan/MINNs
AttentionMIL is implemented at https://github.com/AMLab-Amsterdam/AttentionDeepMIL
KernelSelfAttention-MIL is implemented at https://github.com/gmum/Kernel_SA-AbMILP
MIVAE is implemented at https://github.com/WeijiaZhang24/MIVAE.