# OpenReview forum: "Multi-Instance Causal Representation Learning for Instance Label Prediction and Out-of-Distribution Generalization"
_NeurIPS.cc/2022/Conference — NeurIPS 2022 Accept_

### Official Review · Reviewer_A4sn · 2022-07-07

**Rating:** 4
**Confidence:** 4
**Soundness:** 3 good
**Presentation:** 2 fair
**Contribution:** 2 fair

**Summary:**

This paper proposes a causal representation learning algorithm for multi-instance learning, called TargetedMIL, which aims to identify invariant causal representations of instances from bag-level weak supervision. TargetedMIL separates each instance into a causal and a non-causal parts and estimates labels for each instance. Experiments on synthetic and real-world datasets demonstrate the effectiveness of the proposed algorithm for two tasks.

**Questions:**

How to ensure conditions (b) and (c) in Assumption 1 always hold?

**Limitations:**

My major concern is about the reasonability of the proposed algorithms. The conditions in assumptions used in this paper seems very strict that may be violate in real world. Furthermore, the latent factors identified by the proposed algorithm are hard to evaluate which may violate the assumptions.

**Strengths And Weaknesses:**

Strengths:

1.	This work is well-motivated with theoretical analyses.

2.	The assumptions used are described in detail.

3.	The experiments are conducted on real-world datasets the empirical results outperform some classical methods.

Weaknesses:

1.	The conditions (b) and (c) in Assumption 1 may not hold in real-world applications.

2.	Lack of theoretical analyses about estimation errors using Equations (7) and (8), while Theorem 1 only holds in ideal environment.

3.	Lack of analysis of computational complexity.

4.	The relationship between the two tasks (instance label prediction and OOD Generalization) is unclear.

---

> ### Author Response · Authors · 2022-07-31
> **Thanks for your comments and question! We provide our response below.**
>
> **Question: How to ensure conditions (b) and (c) in Assumption 1 always hold?**
>
> **A:** Thanks for your question. Condition (b) of Assumption 1 states that the causal graph is a directed acyclic graph (DAG), a standard assumption adopted by almost all work related to causal inference. The DAG assumption generally holds for most machine learning application scenarios [1]. Condition (c) is similar to the stable generating mechanism assumption in [2,3]. However, it is easier to satisfy in ours because TargetedMIL only requires that $p(\mathbf{x}\_{ij}|\mathbf{z}\_{ij})$ is invariant across different bags (whereas under standard supervised learning setting it is required to be invariant across different environments).
>
> You are right that some of these assumptions are difficult to verify; however, rigorous identifiability results cannot be obtained without assumptions. In reality, some of the assumptions can be relaxed. For example, (a) may be relaxed to the collective MIL assumption using the work mentioned by *Reviewer RjbB*; (c) may be relaxed to a weaker version where $p(\mathbf{x}\_{ij}|\mathbf{z}\_{ij})$ is subject to small perturbance. Furthermore, the experiments on histopathology images and text classification (please refer to "Additional experiments (Response to Reviewer WkwK, RjbB, A4sn.”) validate the effectiveness of TargetedMIL in real-world applications.
>
> [1] J. Pearl. Causality. Cambridge University Press, 2009.
>
> [2] B. Schölkopf, et al. Toward causal representation learning. Proceedings of the IEEE, 109(5):612–634, 2021.
>
> [3] P. Cui and S. Athey. Stable learning establishes some common ground between causal inference and machine learning. Nature Machine Intelligence, 4(2):110–115, 2022.
>
> **Comment 1: The conditions (b) and (c) in Assumption 1 may not hold in real-world applications.**
>
> **R1:** Please refer to the response above.
>
> **Comment 2: Lack of theoretical analyses about estimation errors using Equations (7) and (8), while Theorem 1 only holds in ideal environment.**
>
> **R2:** Thanks for this comment. Firstly, Eq (7) can represent any permutation-invariant set function [1]. Secondly, for Eq (8), the approximation error comes from the fact that VAE-based methods conduct amortized inference, and the ELBO is a lower bound of the log-likelihood. To our knowledge, no consensus exists regarding the approximation errors in VAE models. However, many empirical results have shown that VAE-based models perform well in a wide range of applications, e.g., vision tasks such as image classification and face recognition, and biology tasks such as protein structure and binding prediction.
>
> We now discuss the assumptions in Theorem 1. The requirement of *assumptions (i) and (iii)* are technical: (i) ensures that $\phi_\varepsilon$ is non-zero, and (iii) guarantees that the Jacobian of $T_{\mathbf{f}}$ exists and has full column rank. *Assumption (ii)* is necessary as if $\mathbf{f}$ is not injective, some information of $\mathbf{z}$ would be unrecoverable. The above three assumptions are required by most of the identifiability results. Furthermore, *assumption (iv)* is quite easily satisfied in MIL. (iv) can be stated as $(\mathbf{\lambda}(\mathbf{B}_1, y) - \mathbf{\lambda}(\mathbf{B}_0, y)$, $\cdots$, $\mathbf{\lambda}(\mathbf{B}_k, y) - \mathbf{\lambda}(\mathbf{B}_0, y))$ are independent. Furthermore, $y$ is unnecessary if there exist $k+1$ distinct bags such that
> $(\mathbf{\lambda}(\mathbf{B}_1) - \mathbf{\lambda}(\mathbf{B}_0)$,$\cdots$,  $\mathbf{\lambda}(\mathbf{B}_k) - \mathbf{\lambda}(\mathbf{B}_0))$ are independent.
>
> Because bags are independent (the instances within bags do not need to be), (iv) can be easily satisfied.
>
> **Comment 3: Lack of analysis of computational complexity.**
>
> **R3:** We did not conduct a computational complexity analysis because the exact cost of TargetedMIL depends on the encoder-decoder networks required for the task. However, a major computational advantage of TargetedMIL over existing MIL algorithms is that TargetedMIL can be trained by a mini-batch containing multiple bags. In contrast, other deep MIL approaches, such as Attention-based MIL, must be trained by mini-batch containing only one bag.
>
> **Comment 4: The relationship between the two tasks (instance label prediction and OOD Generalization) is unclear.**
>
> **R4:** They do seem unrelated as instance label prediction is a multi-instance learning task while OOD generalization is a supervised learning task. TargetedMIL excels on both tasks because it identifies the causal factor $\mathbf{z}^c$. For instance label prediction, TargetedMIL performs significantly better than existing MIL algorithms because focusing on $\mathbf{z}^c$ and excluding $\mathbf{z}^e$ makes it easier to infer instance label from bag supervision. For OOD generalization, TargetedMIL performs well because the causal factor $\mathbf{z}^c$ is invariant across distribution changes and because TargetedMIL accurately predicts instance labels.

---

> > ### Comment · Reviewer_A4sn · 2022-08-08
> > **Comments after author rebuttal**
> >
> > Thanks for your responses.
> >
> > I've read the other reviews and authors' response. But I respectfully disagree with your responses about the assumptions. Specifically, although DAG is a common assumption in causal inference, it is hard to guarantee the correctness of the path direction between the generated latent factors z and the other factors (as well as their invariant generation mechanisms) during the learning process, making it less convincing about the theoretical and empirical results. Besides, two unrelated tasks in this paper make the readers confused about the goal of designing the novel causal-based learning method. My concern about the time efficiency was also not well addressed, which would limit the implementation in practical applications.
> >
> > According the above reasons, I would still vote for reject for the paper.

---

> > > ### Author Response · Authors · 2022-08-08
> > > **Thank you very much for the discussion!**
> > >
> > > Below we further discuss the raised concerns:
> > >
> > > **Guaranteeing the correctness of the path directions**
> > >
> > > This is an interesting problem that is important for causality-based learning algorithms. In the following, we explain our reasoning and argument regarding why it is necessary to have these assumptions even if they are theoretically often unverifiable, and discuss the reasons why they work well empirically.
> > >
> > > Theoretically, every causal inference or causality-based machine learning algorithm must depend on certain unverifiable assumptions [1]. For example, causal structure learning algorithms assume three unverifiable assumptions: Markovian, faithfulness, and causal sufficiency assumptions; causal effect estimation algorithms often assume a known DAG if they build upon do-calculus, or assume unconfoundedness, overlap, and stable unit assumptions if they follow the potential outcome framework. In the same sense, the *DAG assumption* and the *invariant generation mechanism assumption* are the two necessary assumptions adopted by TargetedMIL. Therefore, we argue that although conditions (b) and (c) are unverifiable, relying upon these widely-accepted assumptions is not really a weakness of the causality-based TargetedMIL algorithm.
> > >
> > > Empirically, there are much evidence that a VAE-based model works sufficiently well in learning generative models by maximizing the evidence lower bound (ELBO) derived from its underlying generative process. Our experiment results also support that TargetedMIL effectively identifies the causal factors. Because TargetedMIL identifies the latent causal factor $\mathbf{z}^c$, it empirically outperforms MIL algorithms in instance label prediction tasks and performs better than supervised learning algorithms in OOD generalization tasks. Furthermore, as demonstrated by the recent successes of causality-based learning algorithms that build upon VAE [2,3,4], maximizing the evidence lower bound has also been shown to be an effective way of learning causal representations.
> > >
> > > [1] Judea Pearl. Causality. Cambridge University Press. 2009.
> > >
> > > [2] Yu Yao, Tongliang Liu, Mingming Gong, Bo Han, Gang Niu, Kun Zhang. Instance-dependent Label-noise Learning under a Structural Causal Model. NeurIPS 2021.
> > >
> > > [3] Mengyue Yang, Furui Liu, Zhitang Chen, Xinwei Shen, Jianye Hao, Jun Wang. CausalVAE: Disentangled Representation Learning via Neural Structural Causal Models. CVPR 2021.
> > >
> > > [4] Chaochao Lu, Yuhuai Wu, José Miguel Hernández-Lobato, Bernhard Schölkopf. Invariant Causal Representation Learning for Out-of-Distribution Generalization. ICLR 2022.
> > >
> > > **Relationship between instance label prediction and OOD generalization tasks**
> > >
> > > The motivation for designing the causality-based TargetedMIL algorithms is that we want to utilize the bag information in MIL to learn causal representation for the instances. By achieving this goal, TargetedMIL naturally performs well on both instance label prediction (because the causal representation excludes transformations that are difficult to learn under weak supervision) and OOD generalization tasks (because the causal representation is invariant across distributions).
> > >
> > > We believe that an algorithm that performs well on two tasks should not be viewed as its weakness, but rather a strength that validates the motivation and the effectiveness of TargetedMIL.
> > >
> > > **Computational complexity**
> > >
> > > Thanks for raising the concern regarding the computational cost. To further address this, we empirically evaluated the computation time of 4 representative deep learning-based MIL algorithms, including TargetedMIL, MIVAE, miNet, and AttnMIL. The training time is obtained from running 200 epochs on the ColonCancer dataset. To ensure a fair comparison, all feedforward algorithms (mi-Net, AttnMIL, KSA-MIL) used the same convolutional network structure, and encoder-decoder methods (MIVAE and TargetedMIL) use the same convolutional encoder and a corresponding deconvolutional decoder. The experiments are conducted using a single Nvidia RTX 3090 GPU.
> > >
> > > |             | FLOPs (M) | Training time (s) |
> > > |-------------|-----------|-------------------|
> > > | mi-Net      | 691       | 289.94            |
> > > | AttnMIL     | 704       | 295.72            |
> > > | KSA-MIL     | 788       | 323.64            |
> > > | MIVAE       | 2046      | 750.10            |
> > > | TargetedMIL | 1139      | 446.81            |
> > >
> > > From the results, we can see that the computation time of TargetedMIL is not significantly different to the rest of the deep MIL algorithms. Feedforward algorithms are slightly faster because they do not have a decoder and do not sample from distributions. TargetedMIL is faster than MIVAE because it only explicitly models one latent variable. However, their running times are all of the same magnitude. We hope this comparison could address the reviewer’s concern about the computational complexity.

---

### Official Review · Reviewer_s6pT · 2022-07-08

**Rating:** 8
**Confidence:** 5
**Soundness:** 3 good
**Presentation:** 3 good
**Contribution:** 3 good

**Summary:**

This paper synergizes identifiable variational autoencoder with multi-instance learning, an important weakly supervised learning problem. By utilizing the instances in the multi-instance bags as auxiliary information, the proposed method provably identifies the latent factors of the positive instances up to affine transformations. The proposed method then utilizes the inferred latents and achieves significantly better results in downward tasks such as instance label prediction and out-of-distribution generalization. Empirical evaluations of qualitative latent reconstructions support the identifiability claim. Quantitative results on several benchmark datasets also show that the proposed method is significantly better at predicting instance labels and out-of-distribution generalization than the baselines.

**Questions:**

1.	What is the scope of domains for the proposed causal graph in Figure 1? Is it applicable to weakly-supervised image classification problems? Discussing some practical problems for which this causal graph is suitable would be preferable.
2.	At the high level, how does the proposed VAE-based MIL method compare to the methods that are based mainly on attention, such as [16] and its follow-up works? As VAE-based MIL algorithms is very different from the current trend of attention-based MIL algorithms, what are the considerations when choosing one over another?


**Limitations:**

Yes. The authors have discussed that the method only applies to the standard multi-instance assumption. Solving the identifiability problem in other MIL assumptions remains to be explored.



**Strengths And Weaknesses:**

Strengths:
--	This paper presents a novel method for integrating multi-instance learning with identifiable latent representation learning.
--	The paper is well written and nicely presented.
--	The qualitative latent reconstruction results are novel and interesting, and the quantitative results show significant improvement over the baselines.
--	The out-of-distribution generalizability brought by the identified latent factors broadens the impact of MIL methods to other machine learning subfields.

Overall, the strength of this paper is in the formulation of multi-instance learning as an identifiable VAE with auxiliary information problems. This provides a novel perspective for MIL band motivates better methods not only for instance label prediction but also for out-of-distribution generalization. The focus on multi-instance learning is important as it has been much overlooked compared to how prevalent it is in important real-world applications such as whole-slide medical imaging and fine-grained prediction, and the focus on identifiability has been shown to be useful to further the MIL methodology. In terms of writing, it nicely addresses the challenges of MIL and identifiability and builds up its case coherently. Identifiability is first analyzed under the correct graphical model, then relaxed to accommodate the multi-instance assumption. The model assumptions are clearly stated as well as the assumptions about the data generating process.

Weakness:
--	Some minor issues in the text could be improved with further proofreading. Please see the detailed comments below.

Minor Points:
Line 147, “In Equation 2”, -> “In Equation 2,”
Line 160, “can then be written as” -> “can be written as”
Line 226, “and thus ensures identifiability” -> “and to ensure identifiability”
Line 263, “Appendix” -> “Appendices”
In Table 2, “TargetedMIL” should be in bold font.
Line 628 (in the Appendix), “Table ??” should be “Table 3”.

---

> ### Author Response · Authors · 2022-07-31
> **Thanks for your encouraging feedback! We answer your questions below.**
>
> **Q1: What is the scope of domains for the proposed causal graph in Figure 1? Is it applicable to weakly-supervised image classification problems? Discussing some practical problems for which this causal graph is suitable would be preferable.**
>
> **A1:** The causal graph in Figure 1 is suitable for a wide range of weakly supervised tasks where *the bag labels are determined by the labels of their instances*, such as sound event detection, object detection, and medical image analysis. For example, in histopathology medical image analysis, a whole-slide image is represented by a bag, and the cells are represented by instances. Supervision is only available at the image level, while whether a patch is cancerous or normal is unknown; however, patch level predictions are crucial for interpretability in medical applications. TargetedMIL is suitable because it accurately predicts instance labels by identifying the underlying causal factors of the cancerous cells.
>
> **Q2: At the high level, how does the proposed VAE-based MIL method compare to the methods that are based mainly on attention, such as [16] and its follow-up works? As VAE-based MIL algorithms is very different from the current trend of attention-based MIL algorithms, what are the considerations when choosing one over another?**
>
> **A2:** [16] utilizes the attention mechanism in a feedforward network to aggregate each instance's contribution to the bag label. Because the attention mechanism assigns continuous weights to both positive and negative instances in positive bags, it is not best suited for instance label prediction under the standard multi-instance assumption.
>
> The proposed TargetedMIL algorithm integrates max-pooling with the evidence lower bound to learn an encoder-decoder model with identifiable causal representations, and the identified causal representation makes instance label prediction easier while benefiting model robustness.
>
> In summary, our proposed algorithm should be preferred when the task is instance label prediction, or distribution change exists. Attention-based MIL algorithms are more suitable for bag classification tasks where the training and test datasets follow the same distribution.
>
> **Comment regarding minor text improvements.**
>
> **A:** Thanks for helping us improve the paper and correcting the typos. We addressed these issues and will further proofread the manuscript. All discussed changes will be included in the revised manuscript.

---

> > ### Comment · Reviewer_s6pT · 2022-08-08
> > **Review to comments**
> >
> > I think the idea of using the bag information to prove the identifiability of the latent factors is quite novel and promising. This idea provides a new direction for multi-instance learning research, and the essence of this idea could also work in other weakly supervised learning settings, which may further broaden the impact of this work.
> >
> > The discussions in the rebuttals provide helpful intuitions for understanding the identifiability assumptions of Theorem 1 under the multi-instance learning setting, and the result on classical MIL datasets is also a nice addition. On comments regarding conditions (b) and (c) in Assumption 1 (*raised by Reviewer A4sn*), my understanding is that these conditions are ubiquitously adopted in causal inference and causal representation learning literature. Therefore, it should not be a concern about the reasonability of the algorithm. The authors can add the discussions and clarifications to the camera-ready version.
> >
> > I recommend strong acceptance for this paper.

---

> > > ### Author Response · Authors · 2022-08-09
> > > **Thank you!**
> > >
> > > Thanks very much for the constructive comments and recommendation. We will continue revising the manuscript to incorporate the results and discussions.

---

### Official Review · Reviewer_RjbB · 2022-07-11

**Rating:** 7
**Confidence:** 3
**Soundness:** 3 good
**Presentation:** 2 fair
**Contribution:** 3 good

**Summary:**

This work solves multi-instance learning (MIL) via utilizing bag-level weak supervision. Specifically, the authors propose a general graphical model which disentangles each instance as generated from instance-specific factors and bag-inherited factors. The proposed TargetedMIL algorithm is solved by identifiable variational autoencoder (iVAE).

**Questions:**

1. Does this result apply to collective MIL assumption proposed in [1]? The collective assumption assumes that several instances work together to determine the bag label. This assumption is explored in a related work that generalizes Attn-MIL [2]. Hence, could this work generalize to this MIL assumption?

[1] Multiple instance learning: A survey of problem characteristics and applications. Pattern Recognition.
[2] Deep multiple instance selection. Sci. China Inf. Sci.


**Limitations:**

Yes

**Strengths And Weaknesses:**

Pros:
1. The graphical model seems novel and rational. Under the standard assumption of MIL, instance labels are not observed and their labels determine the bag-level label. Hence, decoupling instance labels to both instance-specific factors and bag-level factors seems reasonable.
2. A novel TargetedMIL algorithm is proposed. TargetedMIL takes advantage of permutation invariant set transformation networks to identify the latent factors and learn instance feature representations.
3. The paper organization is well and clear to understand. Theoretical results are provided.

Cons:
1. The utilized datasets are slightly simple. MNIST, FaMNIST, and KuzushijiMNIST are simple datasets that are not so representative to verify the effectiveness of the proposed method. More complex datasets should be explored.

---

> ### Author Response · Authors · 2022-07-31
> **Thank you for the constructive comments! We answer your questions below.**
>
> **Comment: The utilized datasets are slightly simple. MNIST, FaMNIST, and KuzushijiMNIST are simple datasets that are not so representative to verify the effectiveness of the proposed method. More complex datasets should be explored.**
>
> **Response:** Thanks for this constructive comment. Besides the Colon Cancer results reported in the manuscript, we also report experiments with the multi-instance 20 Newsgroup datasets used in [3] to further verify Targeted MIL. *Please refer to "Additional experiments (Response to Reviewer WkwK and RjbB)."*
>
> **Question: Does this result apply to collective MIL assumption proposed in [1]? The collective assumption assumes that several instances work together to determine the bag label. This assumption is explored in a related work that generalizes Attn-MIL [2]. Hence, could this work generalize to this MIL assumption?**
>
> **Answer:** Thanks for bringing up this insightful question and important reference. [2] provides a viable way to extend our work to the collective MIL assumption. It utilizes Gumbel softmax and Gumbel top-k for the standard and collective MIL assumption, respectively. As the Gumbel reparameterization trick is in synergy with VAE-based methods [4], using Gumbel top-k with our proposed algorithm is likely to work with the collective MIL assumption. This would be an interesting direction for future explorations. We will add the discussion and reference to the revised manuscript.
>
> [3]	Z.-H. Zhou, Y.-Y. Sun, and Y.-F. Li. Multi-instance learning by treating instances as non-I.I.D. samples. *ICML 2009*, pages 1249-1256.
>
> [4] 	E. Jang, S. Gu, and B. Poole. Categorical Reparameterization with Gumbel-Softmax. *ICLR 2017*.

---

> > ### Comment · Reviewer_RjbB · 2022-08-03
> > **After reading the rebuttals**
> >
> > I had a chance to read through the authors' rebuttals and other reviewers' comments. The responses have answered my question well, and my concern has been adequately resolved. Overall, I think this is a novel and well-motivated work that explored the synergy between deep generative multi-instance learning and causal inference, an area that deserves more attention. Furthermore, the superior OOD generalization performance shown over standard supervised learning algorithms has the potential to broaden the impact of multi-instance learning research.
> >
> > I increased my score to acceptance.
> >
> > Soundness: 3, good
> > Contribution: 3, good
> > Rating: 7: Accept

---

> > > ### Author Response · Authors · 2022-08-04
> > > **Thank you**
> > >
> > > Thank you for raising your score. We will incorporate the discussed changes in the manuscript.

---

### Official Review · Reviewer_WkwK · 2022-07-11

**Rating:** 5
**Confidence:** 3
**Soundness:** 3 good
**Presentation:** 2 fair
**Contribution:** 2 fair

**Summary:**

The authors investigate a generative model for MIL data where instances are generated from bag level and instance level latent factors. They develop variational auto encoders to model this generative process and show an identifiability condition. Experimental results are presented on some synthetic MI datasets and one real world problem which show advances on the state of the art.


**Questions:**

see above

**Limitations:**

see above

**Strengths And Weaknesses:**


+ Paper attempts to explore the use of causal models in MIL, which is a nice, not well explored direction.
+ The experimental results are very impressive and it seems the approach can work well in the real world domain investigated.

- The paper is dense and hard to read. There are many typos.
- The paper borrows heavily from two prior papers, one on identifiable VAEs and one on a multiple instance VAE. It was not clear to me that the advances made beyond these were significant in algorithmic or theoretical terms. While the identifiability result is nice, it was not clear if it required analytical tools that were novel beyond the known result for iVAE. The distinction between factorized/non-factorized seems minor; it is not clear how significant this is. There was also very little comparison to MIVAE, although the graphical models are extremely similar. It is stated that the reason this approach could do better than MIVAE was identifiability, but why that is and why the MIVAE was not identifiable was not clear to me. In general, I found the discussion and comparison to prior work lacking both in the theory and in the experiments. This is one significant area for improvement.
- It is not clear if the assumptions in theorem 1 make sense for MIL, especially (iv). Not enough intuition is provided to help understand the necessity/sufficiency of the assumptions.
- in Eq (8), where is the "weighting hyper-parameter alpha"?
- "The maximum operator of Equation 8 is fundamentally different from the pooling operators used"---this was not very convincing to me.
- Only one experiment is done with an instance labeled MI datasets. There are many MIL datasets available with instance labels. The authors should do additional experiments with actual MI datasets rather than synthetic data such as digit recognition.
- From the appendix, it is not clear how hyperparameters were chosen. There did not seem to be a common tuning process for all algorithms.
- As mentioned above, the comparison with MIVAE was not convincing to me. The ablation study in the appendix is helpful but the claim "this is caused by not having identifiability" lacks justification, and does not really explain why. Further explanation is needed to explain the differences between these approaches.
- No information is given about causal interpretations on the clinical dataset. Are causal relationships found clinically validated?
-Limitations are not really included except one line about the standard MIL assumption, which is not a sensible limitation in as much as every model must make some such assumption.

To summarize, I found the approach promising, but the paper has significant room for improvement.

---

> ### Author Response · Authors · 2022-07-31
> **Thank you very much for the thorough review and thoughtful comments! Below we respond to your questions and comments. (Part 1)**
>
> **Q1. Regarding typos.**
>
> Response: Thanks for your careful reading! We will continue to proofread the paper. All discussed changes will be incorporated into the revised version.
>
> **Q1: Regarding comparison with MIVAE, the distinction between factorized/non-factorized priors, and why MIVAE is not identifiable.**
>
> **Response:** Thank you for this constructive comment.
>
> *We first summarize the four differences between TargetedMIL and MIVAE:*
>
> (1) The first difference lies in the generative models. MIVAE explicitly infers two latents: an instance-level $\mathbf{z}^I$ specific to each instance and a bag-level $\mathbf{z}^B$ shared by all instances in a bag. TargetedMIL only models one instance-specific latent $\mathbf{z}$ which is then decomposed into causal factors $\mathbf{z}^c$ and non-causal ones $\mathbf{z}^e$. In TargetedMIL, the bag information is used for conditioning the prior latent distribution instead of explicitly modeled as a latent.
>
> (2) The second difference lies in the prior distributions for the latents, which is crucial for model identifiability. MIVAE assumes that the latents follow unconditional Gaussian priors, which is unidentifiable. However, in TargetedMIL, the prior distribution for the latent is a conditional Gaussian that depends on the bag information, allowing for identifiability.
>
> (3) The third difference is how the algorithms utilize supervision. In MIVAE, the bag label is predicted from both instance-level and bag-level factors, which is not best suited for instance label prediction. Because all instances in the same bag share the same bag-level factor, the bag factor is not useful for predicting the labels of individual instances. Furthermore, as the bag-level factor $\mathbf{z}^B$ varies across bags, using $\mathbf{z}^B$ in prediction makes MIVAE susceptible to distribution change. TargetedMIL predicts the bag label using only the causal factor $\mathbf{z}^c$. This reduces the difficulty of predicting instance labels from bag supervision and gives the model out-of-distribution generalization capability because the biases incorporated in $\mathbf{z}^e$ are excluded.
>
> (4) The fourth difference lies between the construction of the bag prior $p(\mathbf{B})$ in TargetedMIL and the learning of bag-factor $\mathbf{z}^{B}$ in MIVAE. In MIVAE, bag information is modeled as means of Gaussians whic has limited expressive power. However, in TargetedMIL, we model the bag information using a permutation-invariant set function parameterized by neural networks with universally approximate set functions.
>
> *Then, we discuss the necessity of obtaining identifiability results from non-factorized prior distributions.*
>
> You are right that our identifiability results mainly build upon the analytical techniques introduced by iVAE. However, we argue that our main contribution is not in theoretically advancing iVAE but rather in synergistically integrating iVAE with MIL: allowing for non-factorized priors is important for MIL because the dependency among instances is a characteristic intrinsic to many MIL problems [1].
>
> *Lastly, we discuss why MIVAE is not identifiable and why identifiability is important for instance label prediction performance and out-of-distribution generalizability.*
>
> The identifiability of the latent variables comes from conditional priors, i.e., the latents $\mathbf{z}$ in the generative model are conditioned on auxiliary information $\mathbf{u}$. Unconditional Gaussians are unidentifiable because there will always exist some transformation that changes the value of $\mathbf{z}$ but not its distribution. For example, applying rotation to a spherical Gaussian distribution $p(\mathbf{z})$ does not change $p(\mathbf{x})$ but changes $p(\mathbf{z}|\mathbf{x})$, and the two distributions are indistinguishable. In MIVAE, the prior distributions of the two latent variables are unconditional: the instance-level factor $\mathbf{z}^B$ and $\mathbf{z}^I$ are independently modeled as unconditional Gaussian distributions; this is the reason why MIVAE is not identifiable.
>
> Identifiability is particularly important for instance label prediction in MIL because the bag-level supervision is weaker than the instance-level prediction task. An algorithm can still learn useful representations without identifiability, and it may perform well when supervision matches the task or training/test distributions remain unchanged. However, the unidentifiable representations often contain noisy transformations and spurious features, making it harder to infer instance labels from bag supervision and more susceptible to distribution change. Identifiability is also attractive for MIL because the auxiliary information can be provided by the readily available multi-instance bags, whereas additional supervision is often required under standard supervised learning settings.
>
> [1] Z.-H. Zhou, Y.-Y. Sun, and Y.-F. Li. Multi-instance learning by treating instances as non-I.I.D. samples. ICML 2009, pages 1249-1256.

---

> ### Author Response · Authors · 2022-07-31
> **Thank you so much for the thorough review and thoughtful comments! Below we respond to your questions and comments. (Part 2)**
>
> **Q3. Intuitions behind assumptions in Theorem 1.**
>
> **A:** Thank you for thequestion. *Intuitively*, the auxiliary information should "break the symmetry" in the space of representations the model could learn. An analogy is inferring an object's shape from its shadow: if we only observe one shadow of the object, it's difficult to know its shape; however, if we observe multiple objects under similar lighting conditions (from a bag of instances), we may identify the lightings; if we observe an object under different lighting conditions (from many bags of instances), we may identify the underlying shape. Another analogy is to consider the bags as H&E stained histopathology images and the instances as image patches: in one bag, we observe cells under the same staining, which provides information regarding the staining; in different bags, we observe cancerous cells under different staining, which makes it possible to infer the causal representations of cancerous cells. Such information is available in MIL but not in a standard learning setting. *Technically*, assumption (iv) can be stated as the vectors $(\mathbf{\lambda}(\mathbf{B}_1, y) - \mathbf{\lambda}(\mathbf{B}_0, y)$, $\cdots$, $\mathbf{\lambda}(\mathbf{B}_k, y) - \mathbf{\lambda}(\mathbf{B}_0, y))$ are independent. Furthermore, $y$ is unnecessary if there exist $k+1$ distinct bags such that
> $(\mathbf{\lambda}(\mathbf{B}_1) - \mathbf{\lambda}(\mathbf{B}_0)$,$\cdots$,  $\mathbf{\lambda}(\mathbf{B}_k) - \mathbf{\lambda}(\mathbf{B}_0))$ are independent.
>
> Because bags are independent (the instances within bags do not need to be), (iv) can be satisfied.
> Assumption (ii) is necessary as if $\mathbf{f}$ is not injective, some information of $\mathbf{z}$ would be unrecoverable from $\mathbf{x}$.
> The necessity of Assumptions (i) and (iii) are technical: (i) ensures that $\phi_\varepsilon$ is non-zero, and (iii) guarantees that the Jacobian of $T_{\mathbf{f}}$ exists and has full column rank.
>
> **Q4. Weighting hyper-parameter alpha in Eq (8)?**
>
> **A:** The $\alpha$ should be in the term as $\alpha \log p_{\mathbf{\omega}}(Y \vert \mathbf{z})$.
>
> **Q5. The maximum operator of Eq (8).**
>
> **A:** We will remove this paragraph to make space for the new discussions.
>
> **Q6. Additional experiments**
>
> **A:** Thanks for this constructive comment. We followed the setting in MIVAE and experimented with the 20 Newsgroup datasets. *Please refer to “Additional experiments (Response to Reviewer WkwK and RjbB)."* We also conduct OOD experiments with MIVAE and report the test results here:
>
> |             | ColoredMNIST | ColoredFashionMNIST |
> |-------------|--------------|---------------------|
> | ERM         | 0.105±.007   | 0.225±.007          |
> | ERM1        | 0.109±.005   | 0.333±.089          |
> | ERM2        | 0.101±.002   | 0.132±.008          |
> | MinMax      | 0.152±.025   | 0.292±.086          |
> | IRM         | 0.628±.096   | 0.534±.194          |
> | IRM GAME    | 0.599±.027   | 0.702±.015          |
> | iCaRL       | 0.688±.007   | 0.617±.360          |
> | MIVAE       | 0.156±.003   | 0.284±.120          |
> | TargetedMIL | 0.925±.004   | 0.866±.009          |
>
> **Q7. Parameter tuning.**
>
> **A:** Thanks for catching this. The parameters are tuned by grid search using the evidence lower bound. Three parameters are involved in the tuning: the learning rate {1e-2, 1e-3, 1e-4}, the weighting parameter $\alpha$ {1,10,100}, and the latent dimensionality {8,16,24,32}. Furthermore, parameters are only tuned per dataset, e.g., experiments of 10 FashionMNIST-bags used the same parameters.
>
> **Q8. Comparison and differences with MIVAE.**
>
> **A**: Please kindly refer to responses to Q1 and Q5.
>
> **Q9. Clinically validating the causal interpretations on the clinical datasets.**
>
> **A:** Thank you for asking. Unfortunately, we do not currently have collaborators to analyze the representations pathologically (the cancerous representations do seem different from the normal ones). We would love to reach out to collaborators in the future.
>
> **Q10. Limitations.**
>
> **A:** Thanks for the comment. We will add new discussion and expand the discussion on standard MIL assumption: (1) our generative model in Figure 1 only applies to the *causal prediction* task, i.e., predicting effect $y$ from cause $\mathbf{z}^c$ as discussed in [2]. The *anti-causal predictions* task, i.e., predicting cause from effect, is currently unexplored and worth investigating in MIL.
> (2) The standard MIL assumption can be viewed as a simplification of the collective MIL assumption, which assumes that the bag label is determined by instances belonging to more than one concept [3]. For complex vision tasks, e.g., classifying the image of *beach* where it must have latent factors corresponding to *water* and *sand*, the current approach is not sufficient.
>
> [2] B. Schölkopf et al. On causal and anticausal learning. ICML 2012, pages 459-466.
>
> [3] X.-C. Li, et al. Deep multiple instance selection. Sci. China Inf. Sci. 64: 130102 (2021).

---

> > ### Comment · Reviewer_WkwK · 2022-08-09
> > **Thanks!**
> >
> > I thank the authors for writing a detailed response! The intuition provided helps clarify the motivation and I suggest the authors consider including something similar in the paper.
> > I appreciate the authors providing additional experimental results, however, (i) that is not the purpose of these discussions. I am reviewing the paper as submitted. (ii) There are better instance labeled datasets available the authors should consider. For example, there are instance labeled versions of SIVAL datasets, as well as birdsong datasets with instances labeled. Here, the bag structure is not artificial and is inherent to the problem.
> > Interpreting the results for real datasets is critical to understanding whether the approach is actually doing what is claimed.
> > Did you tune the hyperparameters for the baseline approaches as well?
> >
> > I have a better insight into the work as a result of your responses, and I agree there is a reasonable contribution. But, there are still weaknesses in novelty over prior work and empirical results and interpretation. Given this, I am not enthusiastic, but I am ok with accepting. Thanks again to the authors for your responses.

---

> > > ### Author Response · Authors · 2022-08-09
> > > **Thanks you again for the constructive feedback and discussion!**
> > >
> > > Thank you very much for reading our responses and raising the score!
> > >
> > > We will certainly incorporate the discussions into the manuscript. And also, thanks very much for the dataset recommendations; we will run experiments with the suggested datasets in the future.
> > >
> > > **Q: Did you tune the hyperparameters for the baseline approaches as well?**
> > >
> > > **A:** Yes, we did tune the baseline approaches. For the results reported in the paper, we extensively tuned parameters using the suggested parameter ranges in their paper and also expanded the search range. For the new results reported in the response, we took the performances reported in the relevant paper (due to the limited rebuttal timeframe).

---

### Author Response · Authors · 2022-08-01
**Additional experiments (Response to Reviewer WkwK, RjbB, and A4sn).**

Dear reviewers,

We follow the setting in the MIVAE paper [1] and report additional experiments on instance label prediction performance using the multi-instance 20 Newsgroup dataset [2].

|                          | miSVM | KI-SVM | GPMIL | DPMIL    | VGPMIL   | MIVAE          | TargetedMIL    |
|--------------------------|-------|--------|-------|----------|----------|----------------|----------------|
| alt.atheism              | 0.53  | 0.68   | 0.44  | 0.67     | 0.70     | 0.745±.030     | **0.803±.021** |
| comp.graphics            | 0.65  | 0.47   | 0.49  | 0.79     | 0.79     | 0.800±.042     | **0.809±.038** |
| comp.os.ms-windows.misc  | 0.42  | 0.38   | 0.36  | 0.51     | 0.52     | **0.548±.038** | 0.545±.035     |
| comp.sys.ibm.pc.hardware | 0.57  | 0.31   | 0.35  | 0.67     | 0.70     | **0.711±.034** | 0.679±.038     |
| comp.sys.mac.hardware    | 0.56  | 0.39   | 0.54  | 0.76     | 0.79     | 0.783±.035     | **0.810±.033** |
| comp.windows.x           | 0.56  | 0.37   | 0.36  | 0.73     | 0.69     | 0.754±.032     | **0.802±.025** |
| misc.forsale             | 0.31  | 0.29   | 0.33  | 0.45     | 0.54     | 0.553±.334     | **0.615±.036** |
| rec.autos                | 0.51  | 0.45   | 0.38  | **0.76** | 0.71     | 0.720±.024     | 0.731±.035     |
| rec.motorcycles          | 0.09  | 0.52   | 0.46  | 0.69     | 0.76     | 0.766±.029     | **0.812±.023** |
| rec.sport.baseball       | 0.18  | 0.52   | 0.38  | 0.74     | 0.76     | 0.764±.036     | **0.802±.028** |
| rec.sport.hockey         | 0.27  | 0.66   | 0.43  | 0.91     | **0.94** | 0.925±.020     | 0.938±.018     |
| sci.crypt                | 0.57  | 0.47   | 0.31  | 0.68     | 0.82     | 0.773±.036     | **0.843±.021** |
| sci.electronics          | 0.83  | 0.42   | 0.71  | 0.90     | 0.92     | **0.928±.020** | 0.901±.029     |
| sci.med                  | 0.37  | 0.55   | 0.32  | 0.73     | 0.73     | 0.745±.025     | **0.800±.026** |
| sci.space                | 0.46  | 0.51   | 0.32  | 0.70     | 0.74     | 0.748±.027     | **0.786±.026** |
| soc.religion.christian   | 0.05  | 0.53   | 0.45  | 0.72     | 0.73     | 0.753±.035     | **0.761±.034** |
| talk.politics.guns       | 0.57  | 0.43   | 0.38  | 0.64     | 0.72     | 0.714±.038     | **0.759±.026** |
| talk.politics.mideast    | 0.77  | 0.60   | 0.46  | 0.80     | 0.87     | 0.840±.020     | **0.884±.021** |
| talk.politics.misc       | 0.61  | 0.50   | 0.29  | 0.60     | 0.64     | 0.650±.044     | **0.750±.036** |
| talk.religion.misc       | 0.08  | 0.32   | 0.32  | 0.51     | 0.49     | 0.525±.035     | **0.673±.031** |

The reported results are the AUC-PR scores on the test sets, results with the highest score are highlighted in bold font. Parameter are selected using the "alt.atheism" dataset, and the same parameters are used for the rest of the datasets.

[1] W. Zhang. Non-I.I.D. multi-instance learning for predicting instance and bag labels using variational autoencoder. IJCAI 2021, pages 3377-3383.

[2] Z.-H. Zhou, Y.-Y. Sun, and Y.-F. Li. Multi-instance learning by treating instances as non-I.I.D. samples. ICML 2009, pages 1249-1256.

---

### Meta-Review · Area_Chair_R88v · 2022-08-23

**Recommendation:** Accept
**Confidence:** Certain

**Metareview:**

The paper studies multiple instance learning (MIL) by treating bags as auxiliary information, aiming to identify invariant causal representations using only bag labels available in the MIL setting. To achieve identifiability, it is assumed that the prior distribution over the instance latent variables belongs to the non-factorized exponential family conditioning on the bags. This allows the disentanglement between the causal and non-causal factors and only the causal ones are supposed to contribute to the instance labels (while the bag-level labels are used in the proposed objective function in Eq. 8 to accommodate the MIL setting).  Experiments are conducted on multiple datasets to demonstrate the instance prediction and out-of-distribution generalization performance of the proposed TargetedMIL algorithm.

The perspective of learning invariant causal representations is new in the context of multiple instance learning. Reviewers have acknowledged this interesting aspect of the proposed work. Authors and reviewers engaged in a detailed discussion and the authors' rebuttal helped to address some major confusions, which further improved the qualify of the paper. The authors are encouraged to more clearly highlight the key difference from two important references in the final version of the paper, including identifiable VAEs and multiple instance VAE, which are relevant to the proposed work. The causal inference related assumption could also be further clarified as suggested by one reviewer.


**Award:**

No

---

### Decision · Program_Chairs · 2022-09-14

Accept